# DIRICHLET-PRIOR SHAPING: GUIDING EXPERT SPECIALIZATION IN UPCYCLED MoEs

## ABSTRACT

Upcycling pre-trained dense models into sparse Mixture-of-Experts (MoEs) efficiently increases model capacity but often suffers from poor expert specialization due to naive weight replication. Our analysis reveals that upcycled MoEs, even with conventional regularization, exhibit low-confidence, weakly differentiated routing, hindering performance. We introduce Dirichlet-Prior Shaping Loss (DPSL), a novel router regularization technique that directly shapes routing probability distributions by matching expert assignments to a target Dirichlet prior. DPSL offers fine-grained control over expert balance and specialization, and enables encoding of inductive biases such as encouraging experts to focus on specific modalities or tasks, without requiring manual intervention; notably, DPSL is a general tool applicable to any module that outputs categorical probability distributions, extending its utility beyond MoE training. Experiments on upcycled MoE vision-language models (with Qwen2, Phi3, Llama3.2 LLM backbones) show DPSL consistently outperforms upcycling strategies and regularization techniques across standard vision-language benchmarks, addressing the critical issue of poor specialization and fostering more adaptive, higher-performing models.

## 1 INTRODUCTION

Recent advances in large language models (LLMs) and multimodal LLMs (MLLMs) have transformed natural language and vision-language tasks. Model scaling drives this success (Kaplan et al., 2020; Hoffmann et al., 2022), enhancing accuracy and unlocking new capabilities, albeit with significant increases in training and inference costs. Sparse Mixture-of-Experts (MoE) architectures offer a solution by increasing model capacity while maintaining computational efficiency, activating only a subset of parameters ("experts") for each input token (Jacobs et al., 1991; Eigen et al., 2013). Concurrently, sparse upcycling offers an efficient training strategy by initializing an MoE with a pre-trained dense model, thereby accelerating convergence and leveraging existing knowledge (Komatsuzaki et al., 2023), particularly effective for instruction-tuning. The combination of MoE architectures and upcycling is particularly well-suited for advancing MLLMs, enabling more capable multimodal systems, without prohibitive computational overhead. Recent efforts like LLaVA-MoE demonstrate this direction, using MoE structures to enhance MLLM efficiency (Lin et al., 2024).

However, sparse upcycling introduces specific challenges in expert specialization. Naively initializing all MoE experts by replicating the dense model's feedforward network (FFN) weights (Komatsuzaki et al., 2023) leads to weight homogeneity, impeding the router's ability to differentiate experts and fully utilize its capacity, resulting in suboptimal performance (Nakamura et al., 2025). Drop-Upcycling (Nakamura et al., 2025) addresses this by partially re-initializing a random subset of parameters within each expert to promote diversity, but its benefits typically emerge only after extensive training, often exceeding practical instruction-tuning budgets. Specialized upcycling methods such as Branch-Train-MiX (BTX) (Sukhbaatar et al., 2024) fine-tune separate model copies on different datasets to create diverse experts, which are then merged into an MoE and further fine-tuned with a learned router. However, BTX may yield experts specialized in suboptimal ways for MoE routing and can miss positive knowledge transfer, leading to inefficiencies and suboptimal convergence. In addition, standard MoE regularization, such as load-balancing loss (Shazeer et al., 2017; Fedus et al., 2022) and z-loss (Zoph et al., 2022) aim to stabilize training and prevent expert collapse, but do not directly induce specialization from identically initialized experts. Hence, they are unable to overcome the specialization challenges in upcycled MoEs, especially under limited data.

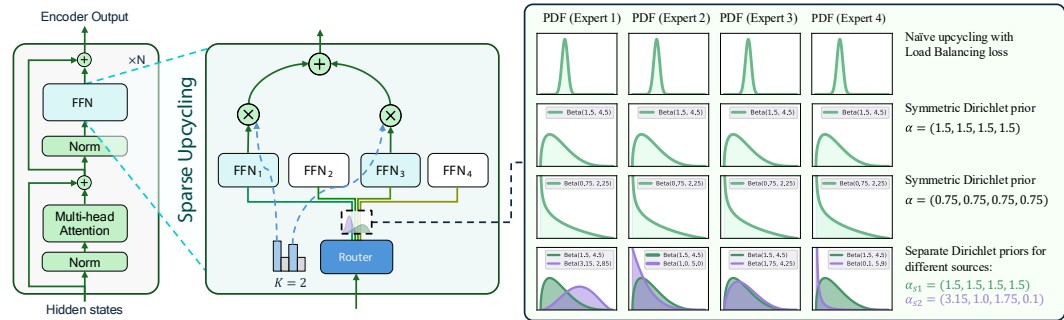

Figure 1: Sparse upcycling (left) initializes identical experts, yielding homogeneous routing probabilities and limited specialization. (right) Our proposed Dirichlet-Prior Shaping Loss guides routing towards desired distributions fostering balanced and confident selection (via symmetric priors) or targeted, modality-/task-aware specialization (via asymmetric priors).

To address the specialization challenges in upcycled MoEs, we first analyze routing behavior and find that, even with conventional regularization, upcycled MoEs exhibit low-confidence, weakly differentiated routing distributions. Expert assignment probabilities remain sharply peaked near $1/N$ (where $N$ is the number of experts), indicating a persistent lack of specialization throughout training.

To overcome this, we propose Dirichlet-Prior Shaping Loss (DPSL), a principled router regularization technique that directly shapes the distribution of routing probabilities using Dirichlet priors (see Figure 1). DPSL generalizes the Batch Shaping Loss (Bejnordi et al., 2020) by matching the empirical distribution of expert assignments to a target Dirichlet prior enabling fine-grained control over both expert balance and specialization. Symmetric priors promote balanced expert utilization, while asymmetric priors allow targeted specialization. In this work, we focus on Vision-Language Models (VLMs), which present novel opportunities for expert specialization in MoEs. In VLMs, the coexistence of distinct modalities and data sources naturally exposes domain structure that MoE routers can harness, creating opportunities for experts to specialize along meaningful axes such as modality, dataset provenance, or task family. By doing so, our framework paves the way for more adaptive and efficient vision-language models. Our main contributions are:

- We analyze the routing dynamics in upcycled MoEs, demonstrating that naive upcycling results in restricted routing probability ranges, and that standard regularization methods fail to effectively promote expert specialization in this setting.
- We propose Dirichlet-Prior Shaping Loss (DPSL), a powerful and flexible tool to instill a wide array of desired statistical properties and behaviors into the learning process of any module that outputs categorical probability distributions. Applied to MoE routers, DPSL enables fine-grained control over expert utilization and specialization.
- We show that asymmetric Dirichlet priors can guide experts towards desired specialization patterns (e.g., modality- or task-specific), without manual intervention or pre-training.
- Through extensive experiments on upcycled MoE variants of state-of-the-art LLMs (Qwen2 (Bai et al., 2023), Phi3 (Abdin et al., 2024), Llama3.2 (Dubey et al., 2024)), we demonstrate that our method significantly outperforms existing upcycling and regularization techniques on standard vision-language understanding benchmarks.

## 2 METHOD

### 2.1 DIRICHLET PRIORS FOR CATEGORICAL DISTRIBUTIONS

Let a model component output a probability vector $\mathbf{p} = [p_1, p_2, \ldots, p_K]$ over $K$ distinct categories, where $\sum_{k=1}^{K} p_k = 1$ and $p_k \geq 0$. As the conjugate prior for categorical distributions, the Dirichlet distribution is the natural choice to model beliefs over such probability vectors. We model $\mathbf{p}$ as drawn from a Dirichlet distribution, $\mathbf{p} \sim \text{Dir}(\boldsymbol{\alpha})$, where $\boldsymbol{\alpha} = [\alpha_1, \ldots, \alpha_K]$ are positive concentration parameters that define the prior. The joint probability density function (PDF) of the Dirichlet distribution is:

$$f(\mathbf{p}; \boldsymbol{\alpha}) = \frac{1}{B(\boldsymbol{\alpha})} \prod_{k=1}^{K} p_k^{\alpha_k - 1}, \tag{1}$$

where the multivariate Beta function is $B(\boldsymbol{\alpha}) = \frac{\prod_{k=1}^{K} \Gamma(\alpha_k)}{\Gamma\left(\sum_{k=1}^{K} \alpha_k\right)}$ and $\Gamma(\cdot)$ is the Gamma function. A key property of the Dirichlet distribution is that each marginal $p_k$ follows a Beta distribution (see Appendix A for derivation): $p_k \sim \text{Beta}(\alpha_k, A - \alpha_k)$, where $A = \sum_{k=1}^{K} \alpha_k$. The Beta distribution with parameters $(\alpha, \beta)$ has the following probability density and cumulative distribution functions:

$$f_{\text{Beta}}(x; \alpha, \beta) = \frac{x^{\alpha-1}(1-x)^{\beta-1}}{B(\alpha, \beta)}; \quad F_{\text{Beta}}(x; \alpha, \beta) = \int_0^x \frac{t^{\alpha-1}(1-t)^{\beta-1}}{B(\alpha, \beta)} dt, \tag{2}$$

where $B(\alpha, \beta) = \frac{\Gamma(\alpha)\Gamma(\beta)}{\Gamma(\alpha+\beta)}$.

The shape of each $p_k$'s distribution depends on both $\alpha_k$ and the total $A$, reflecting dependencies among categories. For symmetric Dirichlet distribution, where all of the elements of the concentration parameter have the same value, larger $\alpha_k$ concentrates $p_k$ near its mean; smaller values yield more dispersed or even U-shaped distributions (see Appendix A.3 for visualizations). By tuning $\boldsymbol{\alpha}$, we can flexibly control the expected distribution over categories: setting all $\alpha_k = 1$ yields a uniform prior, while asymmetric choices (e.g., $\boldsymbol{\alpha} = (\alpha_{\text{high}}, \alpha_{\text{low}}, \ldots)$) bias the distribution toward specific categories. This enables fine-grained control over categorical outputs, as detailed in the next section.

## 2.2 DIRICHLET-PRIOR SHAPING LOSS

To align the empirical distribution of categorical probabilities with a target Dirichlet prior, we adapt the Batch Shaping Loss from Bejnordi et al. (2020), based on the Cramér–von Mises criterion (Anderson, 1962). This criterion measures the squared difference between the empirical cumulative distribution function (CDF), $F_N(x)$, and the target theoretical CDF, $F(x)$:

$$\omega^2 = \int_{-\infty}^{\infty} [F_N(x) - F(x)]^2 dF(x). \tag{3}$$

For each of the $K$ categories, we match the empirical distribution of assigned probabilities $p_k$ (over a batch of samples) to the theoretical Beta distribution, $\text{Beta}(\alpha_k, A - \alpha_k)$, defined by Dirichlet prior.

Let $p_k^{(b)}$ denote the probability assigned to category $k$ for the $b$-th sample in a batch of $B$ total samples. The empirical CDF for the probabilities of category $k$, denoted as $F_N^{(k)}(x)$, is constructed from these probability values. The Dirichlet-Prior Shaping Loss (DPSL), $\mathcal{L}_{\text{DPS}}$, is then computed as the sum of squared differences between the empirical CDF and the target Beta CDF for each category:

$$\mathcal{L}_{\text{DPS}} = \lambda \sum_{k=1}^{K} \frac{1}{B} \sum_{j=1}^{B} \left[ F_N^{(k)}(p_{k,(j)}) - F_{\text{Beta}}(p_{k,(j)}; \alpha_k, A - \alpha_k) \right]^2, \tag{4}$$

where $p_{k,(j)}$ denotes the $j$-th value in the sorted list of probabilities, $p_{k,(1)} \leq p_{k,(2)} \leq \cdots \leq p_{k,(B)}$, assigned to category $k$ across the $B$ samples in the batch. $F_{\text{Beta}}(p; \alpha_k, A - \alpha_k)$ is the theoretical CDF of the Beta distribution with parameters $(\alpha_k, A - \alpha_k)$, and $F_N^{(k)}(p_{k,(j)}) = j/B$ is the value of the empirical CDF at $p_{k,(j)}$. The hyperparameter $\lambda > 0$ controls the strength of this regularization.

Figure 2 illustrates DPSL in practice. For two data sources (S1 in green, S2 in purple) and three output categories, independent Dirichlet priors shape the output distributions. The first two rows show, for each category, empirical CDFs (dashed) and target Beta CDFs (solid) for both sources, at initialization and convergence, respectively; DPSL minimizes the distance between the empirical and target CDFs, thereby encouraging the model's output probabilities for each source to conform to the desired statistical profile. The rightmost bottom plot tracks DPSL convergence during training. The remaining bottom plots show, for each source, the empirical probability histograms per category, overlaid with the target Beta PDFs. For S1, a symmetric Dirichlet prior, $\boldsymbol{\alpha} = (1.5, 1.5, 1.5)$, yields

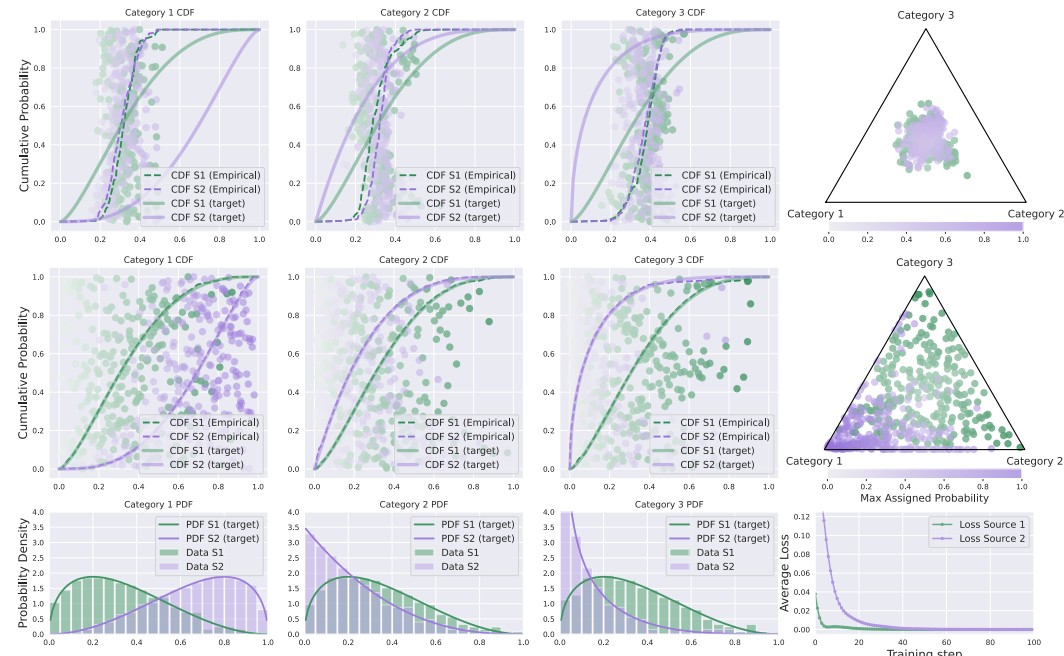

Figure 2: Dirichlet-Prior Shaping Loss (DPSL) shapes categorical probability distributions from two data sources (S1, S2). Top and middle rows show the empirical (dashed) vs. target (solid) CDFs for each category, at initialization and after convergence, respectively, along with simplex of assignment probabilities. Bottom row presents data histograms of assignment probabilities overlaid with target Beta PDFs, and learning curves showing DPSL minimization during training.

balanced probabilities. In contrast, for S2, an asymmetric Dirichlet prior, $\boldsymbol{\alpha} = (3, 1, 0.5)$, induces specialization predominately toward Category one. We provide the training details of this experiment, along with an additional example in Appendix C.1.

In essence, as demonstrated by the example in Figure 2, our Dirichlet-Prior Shaping method offers a powerful and flexible tool to instill a wide array of desired statistical properties and behaviors into the learning process of any module that outputs categorical probability distributions.

## 2.3 DPSL FOR UNSUPERVISED CLUSTERING: A GENERAL APPLICATION

DPSL is fundamentally a general-purpose regularization technique applicable to any module that outputs categorical probability distributions. To demonstrate this broad applicability, we consider a small unsupervised deep clustering problem where a neural network outputs a distribution over $K = 3$ clusters for 2D inputs. As baselines, we adopt SwAV (Caron et al., 2020) and SeCu (Qian, 2023), both of which implicitly promote balanced partitions through an equipartition Sinkhorn–Knopp step (SwAV) or a global entropy constraint (SeCu). On top of these methods, DPSL is added as an auxiliary loss on the cluster assignment probabilities to encode asymmetric Dirichlet priors on cluster sizes, without modifying the underlying clustering objective.

Table 1: Impact of DPSL on unsupervised clustering accuracy.

| Setting | SwAV | SwAV+DPSL | SeCu | SeCu+DPSL |
|---|---|---|---|---|
| Non-overlapping | $84.42 \pm 0.83\%$ | $\mathbf{99.35 \pm 0.14\%}$ (+14.9%) | $88.40 \pm 11.4\%$ | $\mathbf{94.31 \pm 5.49\%}$ (+5.91%) |
| Overlapping | $86.71 \pm 1.13\%$ | $\mathbf{94.09 \pm 0.17\%}$ (+7.4%) | $70.69 \pm 3.8\%$ | $\mathbf{83.28 \pm 1.69\%}$ (+12.6%) |
| Elongated | $87.73 \pm 0.67\%$ | $\mathbf{92.28 \pm 0.46\%}$ (+4.6%) | $76.84 \pm 7.1\%$ | $\mathbf{87.66 \pm 3.86\%}$ (+10.8%) |

We evaluate three challenging synthetic 2D regimes with three clusters and 1500 points per setting: (i) non-overlapping imbalanced clusters with size ratio 4:1:1, (ii) overlapping imbalanced clusters with ratio 5:2:1, and (iii) overlapping elongated clusters with ratio 5:3:1.

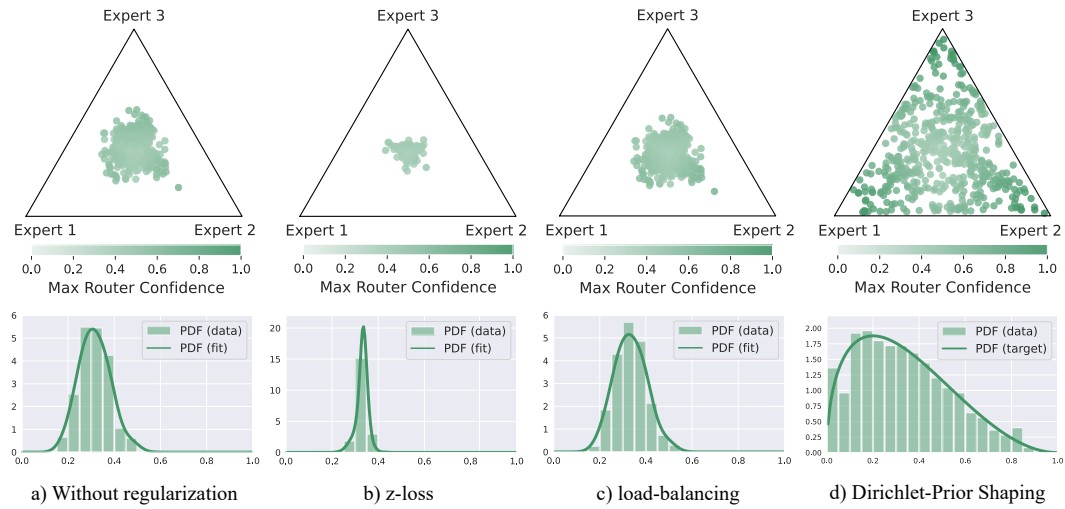

Figure 3: Router output distributions for three experts in an upcycled MoE with top-1 routing. Each panel shows the simplex of routing probabilities under (a) no regularization, (b) z-loss, (c) load-balancing loss, and (d) Dirichlet-Prior Shaping Loss (symmetric prior).

To encode only soft prior knowledge about cluster proportions and overlap, we use moderate asymmetric Dirichlet priors. We report clustering accuracy averaged over three random seeds. As can be seen in Table 1, across all three regimes, adding DPSL substantially improves clustering accuracy for both SwAV (by 4.6%–14.9%) and SeCu (by 5.9%–12.6%), demonstrating that shaping categorical output distributions with Dirichlet priors is a beneficial and general-purpose technique (Qualitative clustering results are presented in Figure 5, Appendix B). Complete experimental details, prior selection rationale, and robustness analysis to prior misspecification are provided in Appendix B.

### 2.4 MOTIVATIONAL STUDY: UNDERSTANDING ROUTER BEHAVIOR IN UPCYCLED MoEs

To motivate the need for a more nuanced control over router learning, especially in the context of upcycled MoE models, we first briefly review MoE fundamentals and then present an empirical study of router output distributions under various common regularization schemes.

#### 2.4.1 MIXTURE-OF-EXPERTS BACKGROUND

Mixture-of-Experts (MoE) architectures increase model capacity and efficiency by activating only a subset of specialized subnetworks, or "experts", for each input token. Each MoE layer replaces a standard feed-forward network (FFN) with $N$ expert FFNs ($E_1, E_2, \ldots, E_N$) and a router module that assigns tokens to experts (see Figure 1 for an example with 4 experts and top-2 routing).

Given a token representation $x$, the router (with weights $\mathbf{W}_g$) computes logits $\mathbf{x}\mathbf{W}_g$, which are converted to routing probabilities $\mathbf{g}(\mathbf{x}) = \mathrm{softmax}(\mathbf{x}\mathbf{W}_g)$. Sparse MoEs typically employ top-$K$ gating: only the $K$ experts with the highest routing probabilities $\mathbf{g}_i(\mathbf{x})$ are selected to process the token. Let $\mathcal{T}_k(\mathbf{x})$ be the set of indices corresponding to these top-$K$ experts for token $\mathbf{x}$. The MoE output is:

$$\mathbf{y}(\mathbf{x}) = \sum_{i \in \mathcal{T}_k(\mathbf{x})} \mathbf{g}_i(\mathbf{x}) \cdot E_i(\mathbf{x}). \tag{5}$$

Recent MoE designs introduce "shared experts" (Dai et al., 2024): in addition to routed experts, shared experts $E_s(\mathbf{x})$, processes all input tokens, akin to a standard FFN. Throughout this paper, we employ MoE architectures with shared experts.

#### 2.4.2 ANALYZING ROUTER OUTPUT DISTRIBUTIONS IN UPCYCLED MoEs

Upcycling a pre-trained dense model into an MoE creates challenges for the router: all experts start as identical FFNs, while the router must learn to differentiate token assignments to foster expert

specialization. We analyzed router output distributions in an upcycled MoE with three experts and top-1 routing, comparing the effects of common regularization techniques.

Figure 3 visualizes the router's output probability distribution for a specific MoE layer, plotted on a simplex where each point represents the probabilities $(p_1, p_2, p_3)$ for a given input. Without regularization (a), router outputs cluster near the center, reflecting low confidence and limited expert differentiation. Applying z-loss (b) (Zoph et al., 2022) further compacts the distribution, stabilizing training but reducing the range and specialization of expert assignments. Load-balancing loss (c) (Fedus et al., 2022) distributes load more evenly but neither improves routing confidence nor encourages a wider probability dynamic range; notably, imbalanced load is often less critical in upcycled MoE training.

In contrast, our proposed Dirichlet-Prior Shaping Loss, illustrated in (d) with a symmetric prior ($\alpha_k = 1.5$), explicitly shapes the router's output distribution, allowing confident and diverse expert assignments while utilizing the full probability range. By choosing appropriate Dirichlet priors, we can flexibly encourage distributions that are confidently skewed or evenly spread as needed, unlike the low-confidence regime of conventional methods.

## 3    EXPERIMENTS AND RESULTS

This section evaluates our proposed Dirichlet-Prior Shaping Loss for training upcycled VLM MoEs. We base our study on the LLaVA framework (Liu et al., 2024b). For the language modeling backbone, we selected Qwen2-1.5B (Bai et al., 2023)), Phi3-mini 3.8B (Abdin et al., 2024), and Llama3.2-1B (Dubey et al., 2024) due to their strong performance and manageable size. Following the setup outlined in LLaVA (Liu et al., 2024b) and MoE-LLaVA (Lin et al., 2024), we utilize CLIP Large (Radford et al., 2021) as the visual encoder.

In the following, we first provide training and implementation details in Section 3.1, then describe the baselines and present downstream evaluation tasks and results in Section 3.2 and Section 3.3. We compare our method to modality- and task-specialized upcycling methods in Section 3.4. Finally, we present ablation studies on the impact of the DPSL's hyperparameters on model performance in Section 3.5 and examine router output distributions and expert specialization patterns in Section 3.6.

### 3.1    TRAINING STAGES AND IMPLEMENTATION DETAILS

We upcycle pre-trained LLMs within the LLaVA framework into MoE architectures, while keeping the vision encoder intact. We investigate two primary MoE configurations: (1) a standard MoE, where each expert is a full FFN replica, and (2) a granular MoE, where each expert is partitioned into multiple smaller ones, allowing more granular experts per token while maintaining constant active parameters (He et al., 2024; Dai et al., 2024; Ludziejewski et al., 2024). The standard configuration corresponds to a granularity of 1, resulting in a 4-expert setup with top-2 routing (2in4). In contrast, the granular MoE configuration uses a granularity of 4, yielding 16 experts (each ¼ the size of a full FFN) with top-8 routing (8in16). Despite the increased number of experts, the total and activated parameter count remains constant across configurations. We further discuss the details of the implementation of the upcycling of FFNs into granular experts in Appendix C.2.

**Training stages.** The training consists of three stages. Initially, we train the MLP projector to map visual tokens into the LLM's embedding space. The subsequent *warm-up stage* aims to bolster the model's general visual-language understanding using a large corpus, predominantly captioning datasets. This stage comprises two phases: first, the dense model with the aligned projector is fine-tuned; second, we introduce the MoE experts and fine-tune the complete MoE architecture, including the experts, router, and other existing parameters. The final *finetuning stage*, involves training on diverse task-specific datasets. This stage aims to refine the experts' capabilities, enabling them to learn the nuances and intricacies of specific tasks. The detailed breakdown of the datasets used in every stage can be found in Appendix C.3 along with implementation details in Appendix C.4. We maintain the same training pipeline and stages for all baselines and our method. Finally, we provide a profiling of the computational overhead introduced by DPSL in Appendix C.5.

**Dirichlet-Prior Shaping Loss for Upcycled MoE Training.** Dirichlet-Prior Shaping Loss (DPSL) is computed at the token level across the entire batch, resulting in an effective sample size of $B = S \times T$, where $S$ denotes the number of sequences and $T$ represents the average sequence length. For each

MoE layer, let $g_i^{(t)}$ be the router's output probability for expert $i$ (among $N$ experts) for the $t$-th token in the batch. DPSL is applied to each router as defined in Equation 4. In our setup, with $S = 128$ and $T$ ranging from 576 to 1024, the effective batch size exceeds 73,000 tokens. We apply DPSL and other router regularization baselines only in the second phase of *warm-up stage*.

## 3.2 BASELINES AND DOWNSTREAM EVALUATIONS

We categorize our baselines into two groups. The first comprises upcycling methods without explicit regularizers: Sparse Upcycling (Komatsuzaki et al., 2023), which directly copies dense model weights to intialize experts, and Drop-Upcycling (Nakamura et al., 2025), which introduces partial weight re-initialization with random noise. The second group includes methods with additional router regularizations: load-balancing loss (Shazeer et al., 2017; Fedus et al., 2022); z-loss (Zoph et al., 2022), and the loss-free DeepSeek balancing procedure (Wang et al., 2024; Liu et al., 2024a). We describe the hyperparameters of these methods in Appendix C.4. Additionally, in Section 3.4, we extend our comparison to include specialized upcycling techniques for various tasks including Branch-Train-MiX (BTX) (Sukhbaatar et al., 2024) as well as a manual routing strategy involving modality-specific warmup to pre-specialize experts for vision and language tokens. For a fair and rigorous comparison, we fine-tuned these baselines for their strongest possible performance, as detailed in Appendix C.4. Finally, we subjected our dense baselines to the exact same enhanced training protocol as our MoE models which resulted in significantly stronger reference accuracies beyond standard practices used in LLaVA (Liu et al., 2024b) and MoE-LLaVA (Lin et al., 2024).

We evaluate our method across six benchmarks. For VQA-style tasks, models are tested on GQA (Hudson & Manning, 2019), TextVQA (Singh et al., 2019), and VizWiz (Gurari et al., 2018). Instruction-following capabilities are assessed using MME (Fu et al., 2023) (consisting of MME-Perception and MME-Cognition), MM-Vet (Yu et al., 2024) and MMBench (Liu et al., 2025). Due to the constraint on the number of submissions for VizWiz evaluation and our large number of baselines and models, we have evaluated all models on the Test-Dev2024 split.

## 3.3 DOWNSTREAM TASK EVALUATION RESULTS

Table 2 summarizes the downstream evaluation results across all evaluated models and upcycling methods. Standard sparse upcycling without regularization shows minimal performance gains, and in some cases, performs worse than the dense baseline, underscoring the challenge of effective expert specialization in naive upcycling. Our Dirichlet-Prior Shaping approach consistently achieves the highest average performance across all models and MoE configurations, including both the standard 2in4 and granular 8in16 expert settings, while the second-best method is a moving target. This consistency demonstrates the effectiveness of our method in promoting expert specialization and robust downstream performance, regardless of backbone or architecture making DPSL a more reliable and generalizable choice. Among the baselines, DeepSeek balancing and Drop-Upcycling are generally strong performers, but their effectiveness varies by model and architecture. For instance, DeepSeek balancing achieves high scores with Phi-MoE 2in4 and Qwen 8in16, but underperforms on Llama 2in4 and Qwen 2in4. Drop-Upcycling performs robustly across most settings, ranking among the top two for Qwen 2in4, but not consistently leading elsewhere. Overall, these results establish Dirichlet-Prior Shaping as the most consistent and broadly effective upcycling regularization strategy among those evaluated.

## 3.4 MODALITY- AND TASK-SPECIALIZED UPCYCLING

This section evaluates our method against specialized upcycling and expert allocation strategies. All experiments here utilize the Upcycled Llama3.2-1B model with 4 experts and top-2 routing.

**Modality-Specific Expert Specialization.** We compare a manual modality-specific routing baseline, where experts are hard-assigned to vision or language tokens during *warm-up*, with mixing allowed only in *finetuning stage*, to our DPSL approach. For the latter, we use modality-aware priors: $\boldsymbol{\alpha}^{(\text{vision})} = (\alpha_b + \alpha_s, \alpha_b + \alpha_s, \alpha_b, \alpha_b)$ for vision tokens and $\boldsymbol{\alpha}^{(\text{language})} = (\alpha_b, \alpha_b, \alpha_b + \alpha_s, \alpha_b + \alpha_s)$ for language tokens, where $\alpha_b$ denotes the base $\alpha$ value and $\alpha_s$ is the additive term to promote increased specialization for the corresponding experts. This encourages soft, learned modality preferences throughout training. As shown in Table 3, manual specialization yields the lowest performance, likely due to its rigidity and lack of early cross-modal sharing. In contrast, our modality-specific DPSL achieves the best results, even slightly outperforming our symmetric DPSL, highlighting the benefit of flexibly integrated, informed priors for MLLMs, whereas suboptimal manual approaches might

Table 2: Downstream task performance comparison of upcycled VLM MoEs methods across various backbone LLMs and MoE configurations. Highest accuracy is marked in bold, 2nd best is underlined. We also report the average accuracy (unweighted mean across seven metrics) for MME-P and MME-C, the scores were normalized over the maximal possible scores (2000 and 800, respectively).

| Backbone | Config | Setup | TextVQA | GQA | MM-Vet | MME-P | MME-C | VizWiz | MMB | Avg |
|---|---|---|---|---|---|---|---|---|---|---|
| Qwen2-1.5B | 2in4 | Dense | **54.28** | 61.43 | 34.3 | 1442.67 | 266.07 | 38.80 | **66.31** | 36.60 |
| | | Sparse Upcycling | 53.14 | 61.65 | 32.9 | 1418.53 | 296.07 | 39.38 | 65.07 | 36.17 |
| | | Drop-Upcycling | 53.23 | **62.10** | 34.9 | 1389.21 | 287.86 | 46.01 | 65.70 | 37.57 |
| | | Load-balancing | 53.66 | 61.42 | 33.0 | 1412.83 | **298.57** | 41.22 | 64.96 | 36.48 |
| | | Z-loss | 53.80 | 61.81 | **36.3** | 1417.29 | 265.00 | 39.04 | 65.86 | 36.84 |
| | | DeepSeek balancing | 53.30 | 61.68 | 33.2 | 1420.25 | 293.92 | 41.87 | 65.92 | 36.72 |
| | | DPSL | 53.01 | 62.01 | 35.3 | **1459.06** | 289.71 | **49.55** | 66.26 | **38.17** |
| | 8in16 | Sparse Upcycling | 53.74 | 62.01 | 33.8 | 1393.42 | 270.00 | 40.74 | 65.75 | 36.72 |
| | | Drop-Upcycling | 54.16 | 61.80 | 34.1 | 1435.91 | 280.35 | 41.30 | **66.36** | 36.97 |
| | | Load-balancing | 53.93 | **62.10** | **34.6** | 1418.28 | 266.78 | 38.16 | 65.80 | 36.52 |
| | | Z-loss | 53.95 | 61.36 | 29.2 | 1394.94 | 266.79 | 39.48 | 65.86 | 35.84 |
| | | DeepSeek balancing | **54.49** | 61.97 | 32.3 | **1444.88** | **310.71** | **43.70** | 65.74 | 37.04 |
| | | DPSL | 53.32 | 61.86 | 34.0 | 1421.90 | 265.00 | 43.54 | 65.98 | **37.25** |
| Llama3.2-1B | 2in4 | Dense | 51.19 | 60.18 | 30.5 | 1295.99 | **253.93** | 35.81 | 61.71 | 34.19 |
| | | Sparse Upcycling | 51.20 | **61.00** | 30.0 | 1309.71 | 251.43 | 40.81 | 60.31 | 34.85 |
| | | Drop-Upcycling | 50.50 | 60.43 | 29.8 | 1293.02 | 236.78 | **43.00** | 62.61 | 35.33 |
| | | Load-balancing | 49.49 | 59.79 | 31.3 | 1331.86 | 247.14 | 40.54 | 59.30 | 34.48 |
| | | Z-loss (2in4) | 51.00 | 60.67 | 30.7 | 1318.65 | 246.07 | 39.62 | 61.49 | 34.92 |
| | | DeepSeek balancing | 51.50 | 60.64 | 29.5 | 1265.25 | 220.00 | 36.65 | 62.39 | 34.51 |
| | | DPSL | **52.82** | 60.98 | **31.7** | **1334.78** | 253.21 | 42.19 | **62.78** | **35.92** |
| | 8in16 | Sparse Upcycling | 51.47 | 60.78 | 28.5 | 1285.61 | 223.57 | 38.66 | 61.88 | 34.92 |
| | | Drop-Upcycling | 51.75 | 60.70 | 32.0 | **1352.30** | **267.40** | 39.50 | **63.30** | 35.47 |
| | | Load-balancing | 49.80 | 59.97 | 28.1 | 1312.68 | 227.50 | **45.53** | 61.32 | 35.09 |
| | | Z-loss (8in16) | 52.06 | 60.92 | **33.5** | 1340.57 | 246.43 | 38.74 | 62.84 | 35.5 |
| | | DeepSeek balancing | **52.89** | **61.32** | 32.2 | 1321.10 | 228.21 | 38.74 | 63.17 | 35.61 |
| | | DPSL | 52.10 | 61.09 | 29.7 | 1294.50 | 247.86 | 44.03 | 63.23 | **35.87** |
| Phi3-mini 3.8B | 2in4 | Dense | **57.32** | 61.78 | 36.5 | **1491.31** | 301.07 | 44.32 | 66.76 | 38.18 |
| | | Sparse Upcycling | 56.90 | 62.64 | 35.4 | 1440.94 | 333.21 | 44.79 | 71.97 | 38.98 |
| | | Drop-Upcycling | 56.55 | **63.01** | 40.3 | 1451.90 | 333.21 | 42.42 | 72.50 | 39.42 |
| | | Load-balancing | 56.57 | 62.78 | 35.1 | 1449.07 | 322.50 | 43.43 | 73.00 | 38.86 |
| | | Z-loss (2in4) | 56.60 | 62.40 | 40.6 | 1467.90 | 311.40 | 46.10 | 73.10 | 39.99 |
| | | DeepSeek balancing | 56.80 | 62.81 | 41.5 | 1481.90 | **361.40** | 43.10 | **73.80** | 39.89 |
| | | DPSL | 56.73 | 62.47 | **42.4** | 1472.80 | 350.00 | **46.20** | 72.31 | **40.18** |

prematurely dismiss such strategies. Following Section 3.5 results, for both modality-specific and task-specific priors, we set $\alpha_{\mathrm{b}} = 0.75$ and $\alpha_{\mathrm{s}} = 0.5$.

**Task-Specific Expert Specialization.** We compare DPSL to Branch-Train-MiX (BTX) (Sukhbaatar et al., 2024) where experts are pre-specialized by fine-tuning separate dense model copies on different data subsets (details in Appendix C.6) before MoE integration. DPSL, instead, applies data-subset-conditional priors during standard upcycled MoE training: for tokens from subset $\mathcal{M}$ (targeting specialization for expert $E_m$), its prior $\boldsymbol{\alpha}^{(m)}$ has a higher $m$-th component (e.g., $\alpha_m^{(m)} = \alpha_{\mathrm{b}} + \alpha_{\mathrm{s}}$) compared to other priors (e.g., $\alpha_j^{(m)} = \alpha_{\mathrm{b}}, j \neq m$). This encourages expert $E_m$ to focus on domain $\mathcal{M}$ while allowing continuous knowledge sharing. While both task-specific methods outperform the dense baseline (Table 3), they underperform our symmetric DPSL strategy. This suggests that for the defined vision-language modeling data subsets, explicitly enforced task specialization might be less

effective than a more general, symmetrically guided approach, possibly due to multi-faceted nature of VLM tasks, non-optimal data subsets, or over-constraining experts compared to allowing more data-driven specialization.

Table 3: Performance comparison of modality- and task-specific expert specialization strategies on Llama3.2-1B (2in4) performance.

| Model | TextVQA | GQA | MM-Vet | MME-P | MME-C | VizWiz | MMB | Avg |
|---|---|---|---|---|---|---|---|---|
| Dense | 51.19 | 60.18 | 30.5 | 1295.99 | 253.93 | 35.81 | 61.71 | 34.19 |
| DPSL (symmetric-prior) | **52.82** | 60.98 | 31.7 | **1334.78** | 253.21 | 42.19 | 62.78 | 35.92 |
| Manual (modality) | 51.60 | 60.82 | 30.3 | 1323.09 | 242.14 | 37.37 | 61.49 | 34.65 |
| DPSL (modality-prior) | 51.83 | **61.40** | **32.1** | 1304.96 | **285.00** | 42.88 | **64.01** | **36.18** |
| BTX (task) | 50.69 | 60.64 | 31.0 | 1330.64 | 247.50 | 40.22 | 63.62 | 35.31 |
| DPSL (task-prior) | 51.99 | 60.73 | 27.8 | 1301.12 | 238.21 | **43.82** | 62.16 | 35.35 |

## 3.5 ABLATION STUDY

**Concentration parameter.** We ablate the symmetric Dirichlet concentration $\alpha$ to assess sensitivity of DPSL to prior sharpness, where smaller $\alpha$ encourages sparser, corner-biased routing and larger $\alpha$ favors more uniform, center-biased assignments, with $\alpha = 1$ corresponding to the flat Dirichlet over the simplex. We present results and detailed discussion in Appendix D.1 (Table 8 for Llama3.2-1B and Table 9 for Qwen2-1.5B). Taken together, the ablation indicates that Llama3.2-1B benefits from a modestly lower concentration ($\alpha = 0.75$), whereas larger backbones such as Qwen2-1.5B are comparatively robust across a wider range of $\alpha$ values. In practice, we adopt backbone-specific defaults for all experiments: $\alpha = 1$ for Qwen and Phi models, and $\alpha = 0.75$ for Llama.

**Concentration parameter in the specialization setting.** We analyze the concentration parameter within the modality-specialization setup, varying both the number of experts per modality and the prior allocations, and report all results and discussion in Appendix D.2 (Table 10). In brief, DPSL remains stable under small changes to the symmetric base prior, while deliberately unbalanced allocations across modalities materially reduce overall accuracy.

**Regularization weight $\lambda$.** We ablate the regularization weight $\lambda$ of DPSL over the range $\{0.001, 0.01, 0.1\}$ and provide complete results in Table 11 in Appendix D.3. The best result is achieved with $\lambda = 0.01$, which is used as the default for all experiments.

**Practical guidelines for selecting $\alpha$.** We view the Dirichlet concentration $\alpha$ as a semantic prior for the desired routing profile. Based on our experiments, we recommend a symmetric unit prior ($\alpha = 1$) as a robust universal default. This setting inherently provides a balanced loading profile for all experts and yields competitive performance across diverse LLM architectures and transfers zero-shot to completely different domains, such as unsupervised clustering (Section 2.3). We observe that slightly smaller values (e.g., $\alpha \approx 0.75$) can be used on smaller backbones to encourage marginally stronger specialization.

For asymmetric priors, where a specific expert or cluster $k$ is preferred (e.g., due to known modality imbalance or domain importance), we recommend starting from the base value (e.g., 1.0) and adding a moderate scalar bias of $\approx 0.5$ to that component's concentration (e.g., $\alpha = (1.5, 1, 1)$). This approach, which we successfully verified in MoE and unsupervised learning experiments, softly biases probability mass toward the target component without enforcing rigid hard constraints.

## 3.6 ROUTING DISTRIBUTIONS AND EXPERT SPECIALIZATION PATTERNS

We qualitatively analyze the distributions of router scores resulting from training with various regularization techniques, including router z-loss, load-balancing loss, and the loss-free DeepSeek balancing method. Appendix D.4 visualizes these output score distributions for a Llama3.2-1B model

configured with 4 experts and top-2 routing. As can be seen, most conventional methods yield router score distributions that are sharply peaked around the uniform selection probability of 0.25 (i.e., $1/4$ for four experts). This clustering suggests that these approaches often result in low router confidence when selecting among experts. In contrast, the model trained with our DPSL exhibits a noticeably broader and more varied distribution of routing scores.

We further examine the expert specialization patterns by computing pairwise cosine similarity between expert activations across layers, see Appendix D.6 for detailed analysis for LLama3.2-1B 2in4. Our DPSL maintains the lowest average similarity compared to load-balancing loss and z-loss, demonstrating superior expert specialization.

We additionally analyze the expert utilization by measuring the Coefficient of Variation (CoV) of expert loads at different layers. We present our finding in the Appendix D.5 (Table 12) for Llama3.2-1B 2in4 model. We can observe that even without explicit enforcing of expert balancing, DPSL loss intrinsically encourages a balanced load distribution consistently visible across layers.

## 4 RELATED WORK

Our work builds upon advancements in upcycling pre-trained dense models into MoE architectures, a technique to efficiently enhance model capacity (Komatsuzaki et al., 2023; Lin et al., 2024). Naive sparse upcycling typically involves replicating feed-forward network weights, which can lead to initial expert homogeneity and low specialization. To address this, methods like Drop-Upcycling (Nakamura et al., 2025) introduce partial re-initialization to promote expert diversity from the start. Further refinement in upcycling enables the creation of fine-grained MoE architectures, notably through the "virtual group" initialization proposed by He et al. (2024), which we leverage for our granular MoE variants. While these methods focus on initialization, our Dirichlet-Prior Shaping Loss offers a distinct approach by providing continuous, fine-grained control over expert specialization throughout the training process by directly shaping the router's output probability distributions.

Effective MoE training also relies on managing router behavior and expert utilization. Common strategies include load-balancing losses to encourage uniform expert activation (Shazeer et al., 2017; Fedus et al., 2022) and router z-loss to improve training stability by penalizing large logits (Zoph et al., 2022). Entropy-based regularization mechanisms (Chen et al., 2025) are applied per token and push each individual routing distribution toward high entropy, thereby discouraging confident assignments and typically shrinking all router outputs toward the center of the simplex. More recent developments include auxiliary-loss-free load balancing, such as dynamically adjusting expert-wise biases used in DeepSeek v3 model Wang et al. (2024); Liu et al. (2024a). Unlike these methods that primarily target even load distribution or numerical stability, our DPSL directly models and regularizes the entire categorical distribution of routing probabilities. DPSL exposes explicit, interpretable control knobs via the Dirichlet concentration parameters. By varying the magnitude and asymmetry of the prior, practitioners can smoothly interpolate between uniform routing, confident but balanced specialization, and targeted specialization (e.g., modality-aware or task-aware), without changing the underlying architecture or adding hand-crafted heuristics.

## 5 CONCLUSION

In this paper, we introduce Dirichlet-Prior Shaping Loss (DPSL), a novel and principled regularization technique that empowers fine-grained control over modules outputting categorical probabilities by aligning their empirical distributions with a target Dirichlet prior. Applied to upcycled VLM MoEs, DPSL demonstrates robust, consistently superior performance across diverse models and MoE configurations. Our results further reveal the promise of modality-specific priors for multimodal learning, enabling more adaptive and effective expert allocation in MLLMs. While this work focused on upcycled MoEs, the principles of DPSL extend naturally to training MoEs from scratch and potentially to a wider array of machine learning systems, opening exciting future directions for instilling desired statistical behaviors directly into the learning process. Specifically, we envision DPSL playing a critical role in stabilizing early-stage MoE training by preventing expert collapse through symmetric prior shaping. Despite broad improvements across backbones and MoE configurations, a primary limitation of our work is the inability to perform multiple seeds across the full matrix of backbones, granularities, and priors due to the expense of upcycled MoE training.

# 6 ETHICS AND REPRODUCIBILITY STATEMENTS

We adhere to the ICLR Code of Ethics. This paper focuses on training methodology to enhance MoE upcycling, however, the model itself incorporates an LLM that may perpetuate biases present in the training data, potentially affecting fairness and reliability. Therefore, we recommend adhering to standard ethical guidelines for the use of LLMs to mitigate these risks.

During the preparation of this manuscript, we utilized large language models (LLMs) to assist with grammar correction and refinement of the writing.

In this paper, we provide all the necessary details to ensure the reproducibility of the presented method. We provide the theoretical justification of the method in Section 2 and Appendix A, implementation details and training protocoles in Section 3.1, Appendix C.2 and Appendix C.4, and data description in Appendix C.3.

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

APPENDIX

## A MARGINALS OF THE DIRICHLET DISTRIBUTION

In this appendix, we provide proof that the marginal distribution of each component $p_k$ of a Dirichlet random vector $\mathbf{p} = (p_1, \ldots, p_K) \sim \text{Dir}(\boldsymbol{\alpha})$ follows a Beta distribution. We first establish the aggregation property of the Dirichlet distribution, then use it to derive the marginal.

### A.1 AGGREGATION PROPERTY OF THE DIRICHLET DISTRIBUTION

**Statement:** If $\mathbf{p} = (p_1, \ldots, p_i, \ldots, p_j, \ldots, p_K) \sim \text{Dir}(\alpha_1, \ldots, \alpha_i, \ldots, \alpha_j, \ldots, \alpha_K)$, then the vector $\mathbf{p}'$ obtained by aggregating components $p_i$ and $p_j$ into a single component $p'_i = p_i + p_j$, i.e., $\mathbf{p}' = (p_1, \ldots, p_i + p_j, \ldots, p_{j-1}, p_{j+1}, \ldots, p_K)$, follows a Dirichlet distribution with parameters $(\alpha_1, \ldots, \alpha_i + \alpha_j, \ldots, \alpha_{j-1}, \alpha_{j+1}, \ldots, \alpha_K)$.

**Proof:** Without loss of generality, we aggregate the first two components, $p_1$ and $p_2$. We want to find the marginal distribution of $\mathbf{p}' = (y, p_3, \ldots, p_K)$, where $y = p_1 + p_2$, by integrating the joint PDF of $(p_1, p_2, p_3, \ldots, p_K)$ over the region defined by $p_1 + p_2 = y$, keeping $p_3, \ldots, p_K$ fixed. We integrate with respect to $p_1$, while substituting $p_2 = y - p_1$. Based on Equation (1) in Section 2.1, the PDF for $(y, p_3, \ldots, p_K)$ is:

$$f(y, p_3, \ldots, p_K) = \int_0^y \frac{1}{B(\boldsymbol{\alpha})} p_1^{\alpha_1 - 1} (y - p_1)^{\alpha_2 - 1} \left( \prod_{k=3}^K p_k^{\alpha_k - 1} \right) dp_1 \tag{6}$$

$$= \frac{1}{B(\boldsymbol{\alpha})} \left( \prod_{k=3}^K p_k^{\alpha_k - 1} \right) \int_0^y p_1^{\alpha_1 - 1} (y - p_1)^{\alpha_2 - 1} dp_1 \tag{7}$$

Applying a change of variables $p_1 = yt$, and evaluating the integral:

$$\int_0^y p_1^{\alpha_1 - 1} (y - p_1)^{\alpha_2 - 1} dp_1 = \int_0^1 (yt)^{\alpha_1 - 1} (y - yt)^{\alpha_2 - 1} (y\, dt) \tag{8}$$

$$= y^{\alpha_1 + \alpha_2 - 1} \int_0^1 t^{\alpha_1 - 1} (1 - t)^{\alpha_2 - 1} dt \tag{9}$$

The remaining integral is the definition of the Beta function $B(\alpha_1, \alpha_2)$. Substituting this back in Equation 7:

$$f(y, p_3, \ldots, p_K) = \frac{B(\alpha_1, \alpha_2)}{B(\boldsymbol{\alpha})} y^{\alpha_1 + \alpha_2 - 1} \prod_{k=3}^K p_k^{\alpha_k - 1} \tag{10}$$

The constant term $\frac{B(\alpha_1, \alpha_2)}{B(\boldsymbol{\alpha})}$ is:

$$\frac{B(\alpha_1, \alpha_2)}{B(\boldsymbol{\alpha})} = \frac{\frac{\Gamma(\alpha_1)\Gamma(\alpha_2)}{\Gamma(\alpha_1 + \alpha_2)}}{\frac{\Gamma(\alpha_1)\Gamma(\alpha_2)\Gamma(\alpha_3)\cdots\Gamma(\alpha_K)}{\Gamma(\alpha_1 + \alpha_2 + \alpha_3 + \cdots + \alpha_K)}} = \frac{\Gamma(\alpha_1 + \alpha_2 + \alpha_3 + \cdots + \alpha_K)}{\Gamma(\alpha_1 + \alpha_2)\Gamma(\alpha_3)\cdots\Gamma(\alpha_K)} \tag{11}$$

This is the reciprocal of the multivariate Beta function for parameters $(\alpha_1 + \alpha_2, \alpha_3, \ldots, \alpha_K)$. Let $\boldsymbol{\alpha}' = (\alpha_1 + \alpha_2, \alpha_3, \ldots, \alpha_K)$. Then the constant is $\frac{1}{B(\boldsymbol{\alpha}')}$. So, the PDF becomes:

$$f(y, p_3, \ldots, p_K) = \frac{1}{B(\boldsymbol{\alpha}')} y^{(\alpha_1 + \alpha_2) - 1} \prod_{k=3}^K p_k^{\alpha_k - 1} \tag{12}$$

Therefore, the marginal distribution of $\mathbf{p}'$ is exactly a Dirichlet distribution $\text{Dir}(\alpha_1 + \alpha_2, \alpha_3, \ldots, \alpha_K)$. This proves the aggregation property for summing two components. The argument can be extended by induction to summing any number of components.

## A.2 MARGINALS OF THE DIRICHLET DISTRIBUTION ARE BETA DISTRIBUTIONS

Using the aggregation property proven above, we can derive the marginal distribution of a single component $p_i$. Aggregate all components except $p_i$ into a single component:

$$p_{-i} = 1 - p_i = \sum_{k \neq i} p_k. \tag{13}$$

By the aggregation property, we have:

$$(p_i, p_{-i}) \sim \text{Dir}(\alpha_i, A - \alpha_i), \tag{14}$$

where $A = \sum_{k=1}^{K} \alpha_k$. Since, the 2-dimensional Dirichlet distribution is equivalent to a Beta distribution, it follows that:

$$p_i \sim \text{Beta}(\alpha_i, A - \alpha_i). \tag{15}$$

This proves that the marginals of a Dirichlet distribution are Beta distributed, as stated in Section 2.1.

## A.3 VISUALIZATION OF THE MARGINAL BETA DISTRIBUTIONS

Figure 4 visualizes the marginal Beta distributions for each component of a Dirichlet distribution. For a symmetric Dirichlet distribution, where all of the elements of the concentration parameter have the same value, larger $\alpha_k$ concentrates $p_k$ near its mean (e.g. $\text{Dir}(5.0, 5.0, 5.0)$); smaller values yield more dispersed or even U-shaped distributions (e.g. $\text{Dir}(0.2, 0.2, 0.2)$); while an $\alpha = 1$ known as the flat Dirichlet distribution corresponds to a uniform distribution over the simplex ($\text{Dir}(1.0, 1.0, 1.0)$). Finally, we present the marginal beta distributions when an asymmetric concentration parameter is used ($\text{Dir}(0.75, 0.1, 1.25)$) in which the last component has the biggest value placing more mass at this component.

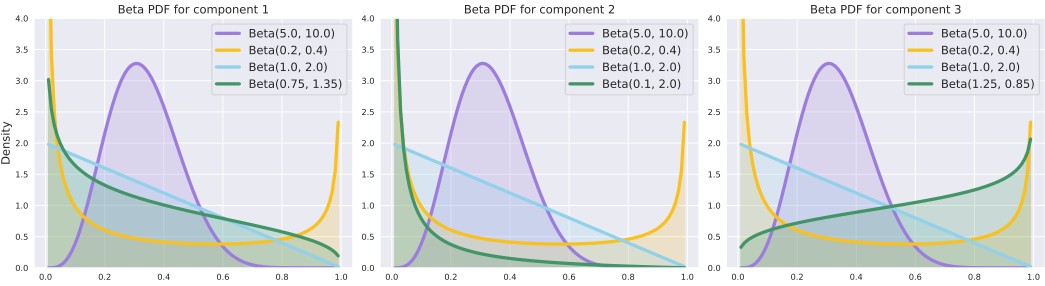

Figure 4: Visualization of the marginal Beta distributions for the following Dirichlet distributions: —$\text{Dir}(5.0, 5.0, 5.0)$, —$\text{Dir}(0.2, 0.2, 0.2)$, —$\text{Dir}(1.0, 1.0, 1.0)$, and —$\text{Dir}(0.75, 0.1, 1.25)$.

# B  PRIOR-GUIDED UNSUPERVISED CLUSTERING EXPERIMENT

To demonstrate the applicability of DPSL beyond MoE routing, we design a synthetic unsupervised clustering task where a small network outputs 3-way cluster-assignment probabilities for 2D inputs. For each of three regimes, we sample 1500 points from a mixture of three Gaussian-like clusters with distinct means and covariances.

## B.1 DATA GENERATION

- **Non-overlapping imbalanced clusters:** The points distributed across three clusters with size ratios 4:1:1, sampled from isotropic Gaussian distributions with standard deviation $\sigma = 0.8$ and fixed centroids $\boldsymbol{\mu}_1 = (0, 0)$, $\boldsymbol{\mu}_2 = (5, 5)$, and $\boldsymbol{\mu}_3 = (-5, 5)$.
- **Overlapping imbalanced clusters:** The data points with size ratios 5:2:1 were sampled from anisotropic Gaussian distributions centered at the vertices of an equilateral triangle (radius 2.4), with varying standard deviations $\sigma \in \{1.0, 0.8, 0.6\}$ and elongation factors $\{1.0, 2.5, 1.5\}$ oriented to induce complex boundary overlaps.

- **Overlapping elongated clusters:** The data points with size ratios 5:3:1 were sampled from highly anisotropic Gaussian distributions centered at the vertices of an equilateral triangle (radius 3.8), with distinct elongations $\{3.0, 4.0, 2.0\}$ and standard deviations $\sigma \in \{0.7, 1.0, 0.5\}$.

### B.2 TRAINING DETAILS

For each setting, we trained an MLP with architecture $2 \to 64 \to 32 \to 3$ with ReLU activations. The final output is passed through a softmax function to produce cluster assignment probabilities. The model is trained for 50 epochs using the Adam optimizer with a learning rate of 0.01 and full-batch gradient descent ($N = 1500$). We use SwAV (Caron et al., 2020) and SeCu (Qian, 2023) unsupervised clustering methods as baselines.

To allow the intrinsic clustering structure to emerge before imposing prior constraints, we implement a warm-up schedule. The model is trained solely with the baseline clustering loss for the first 40 epochs. For the remaining epochs, DPSL is added to the objective with a $\lambda$ set to 0.01 to shape the distribution of the learned cluster assignments towards the target Dirichlet prior.

### B.3 CHOICE OF DIRICHLET PRIORS FOR CLUSTERING

The idea cluster-size ratios and the amount of cluster overlap are generally not known exactly a priori, and even when approximate ratios are available, the effective proportions in the learned representation can deviate due to overlap or elongated cluster geometry. For this reason, we use moderate asymmetric Dirichlet priors that provide soft inductive bias toward approximate proportions without rigid enforcement. In particular, we set $\alpha = (2, 1, 1)$ for the 4:1:1 non-overlapping setting, and $\alpha = (1.5, 1.0, 0.5)$ for both the 5:2:1 overlapping and 5:3:1 elongated settings. The clustering accuracy results for our aligned priors are presented in Table 1 in the main paper.

As qualitatively shown in Figure 5, while the SwAV baseline fails to distinguish smaller or overlapping clusters, adding DPSL effectively guides the model to recover the true cluster structure across all three regimes.

### B.4 ROBUSTNESS TO PRIOR MISSPECIFICATION

To test robustness to prior misspecification, we intentionally used a fixed sub-optimal prior $\alpha = (1.5, 1, 1)$ for all settings which no longer encodes differences between the medium and smallest clusters. Even under these misspecified priors, DPSL consistently improves performance. The results in Table 4 indicate that DPSL remains beneficial even with coarse or partially incorrect prior.

Table 4: Impact of DPSL on unsupervised clustering accuracy, when using a fixed sub-optimal prior for all the settings.

| Setting | SwAV | SwAV+DPSL | SeCu | SeCu+DPSL |
|---------|------|-----------|------|-----------|
| Non-overlapping | $84.42 \pm 0.83\%$ | $\mathbf{88.31 \pm 0.19\%}$ (+3.89%) | $88.40 \pm 11.4\%$ | $\mathbf{93.42 \pm 6.42\%}$ (+5.02%) |
| Overlapping | $86.71 \pm 1.13\%$ | $\mathbf{89.62 \pm 0.30\%}$ (+2.91%) | $70.69 \pm 3.8\%$ | $\mathbf{82.66 \pm 2.83\%}$ (+11.97%) |
| Elongated | $87.73 \pm 0.67\%$ | $\mathbf{88.18 \pm 0.63\%}$ (+0.45%) | $76.84 \pm 7.1\%$ | $\mathbf{83.67 \pm 3.58\%}$ (+6.83%) |

## C TRAINING AND IMPLEMENTATION DETAILS

### C.1 TRAINING DETAILS FOR THE EXPERIMENT IN SECTION 2.2

This appendix provides the training details with an additional illustrative example for applying the Dirichlet-Prior Shaping Loss (DPSL), as referenced in Section 2.2. The objective is to guide a set of learnable probability distributions over three categories to match target Dirichlet priors.

In the example shown in Figure 2 in the paper and Figure 6 in this section, we consider data points representing probability distributions derived from two distinct sources. Independent Dirichlet priors

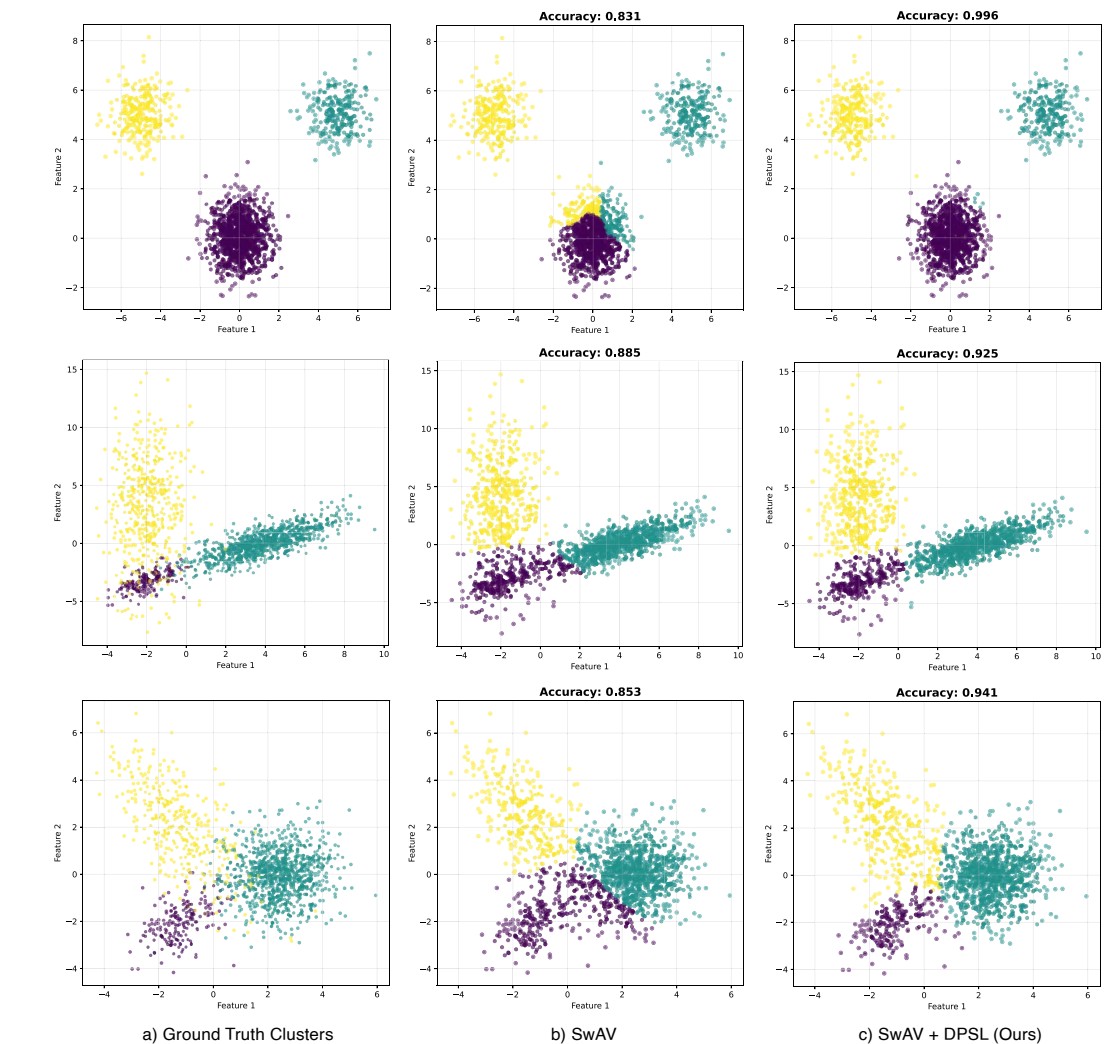

Figure 5: Qualitative impact of DPSL on clustering under various data regimes. We visualize cluster assignments for a random seed across three settings: Top: Non-overlapping (size ratio 4:1:1), Middle: Elongated (ratio 5:3:1 ), and Bottom: Overlapping (ratio 5:2:1). The columns compare the Ground Truth labels (a) with predictions from the SwAV baseline (Caron et al., 2020) (b) and SwAV + DPSL (c). Adding DPSL successfully recovers the distinct cluster structures by shaping the output distribution toward the expected asymmetric prior.

are applied to shape the distributions for each source: For example in Figure 6, Source one has a target prior of $\mathrm{Dir}(5, 5, 5)$ and Source two has a target prior of $\mathrm{Dir}(0.2, 0.2, 0.2)$.

For training, we initialize the data points as learnable parameters. These parameters are optimized using the Adam optimizer with a learning rate of 0.1 for 100 training steps. The optimization minimizes the Dirichlet-Prior Shaping Loss (defined in Equation 4), which quantifies the difference between the empirical CDF of the learned probabilities (for each category) and the theoretical Beta CDF derived from the respective target Dirichlet prior. The learning curve, shown in the bottom right panel of Figure 6, tracks the minimization of this loss during training.

As illustrated in Figure 6, minimizing the CDF divergence ensures that the empirical distribution of the learnable probability vectors for each source converges effectively to its specified target Dirichlet prior (top row). The choice of concentration parameters ($\alpha_k$) significantly influences the characteristics of the learned distributions. For source one, the larger $\alpha_k = 5$ values steer the probability distributions towards the mean of the simplex. For source two, the smaller $\alpha_k = 0.2$

values promote sparse probability distributions. This results in distributions heavily concentrated at the corners of the simplex, where one category is assigned a high probability, and the others are assigned probabilities near zero, indicating a strong preference for a single category.

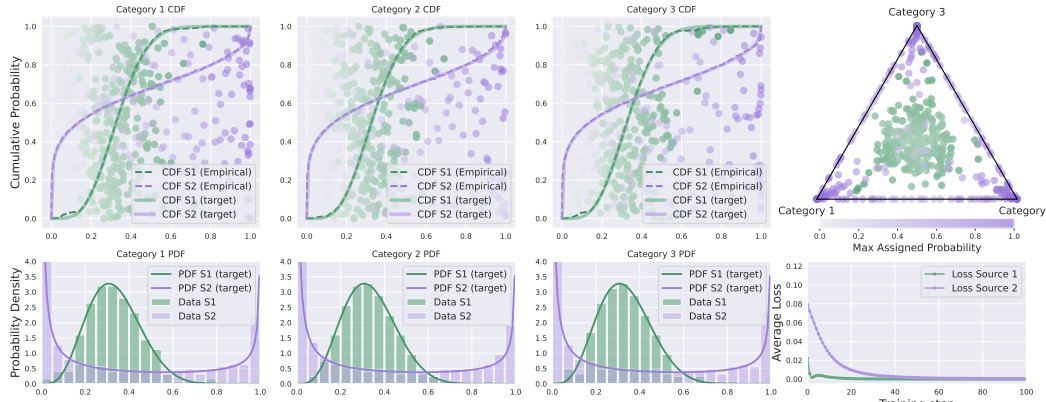

Figure 6: Dirichlet-Prior Shaping Loss (DPSL) shapes categorical probability distributions from two data sources (S1, S2). Top row shows the empirical (dashed) vs. target (solid) CDFs for each category after convergence, along with simplex of assignment probabilities. Bottom row presents data histograms of assignment probabilities overlaid with target Beta PDFs, and learning curves showing DPSL minimization during training.

## C.2 IMPLEMENTATION DETAILS FOR UPCYCLING FFNS INTO GRANULAR EXPERTS

This section provides implementation details for upcycling Feed-Forward Networks (FFNs) into granular experts, as referenced in Section 3.1. Granularity, in this context, refers to the ratio of the original FFN's hidden dimension ($d_{\texttt{ffn}}$) to the hidden dimension of an MoE expert ($d_{\texttt{exp}}$), expressed as $G = d_{\texttt{ffn}}/d_{\texttt{exp}}$. Creating smaller, more granular experts allows tokens to be routed to a larger number of experts, which has shown promising accuracy results (Ludziejewski et al., 2024; He et al., 2024) for granular expert upcycling. We closely follow the approach detailed in He et al. (2024).

We experimented with both standard upcycling and fine-grained upcycling, as follows: For standard upcycling, we duplicate the original FFN blocks to create experts. We add noise to the weights at initialization with a small magnitude, $\epsilon \sim \mathcal{N}(0, 0.01)$. For fine-grained upcycling, we follow the approach proposed by He et al. (2024), partitioning each FFN weight tensor into $G$ shards. In our experiments, upcycling with granularity 1 (standard upcycling) corresponds to a setup with 4 experts and top-2 routing. Fine-grained upcycling corresponds to a setup with 16 experts and top-8 routing.

Notably, we implemented weight scaling for expert initialization (He et al., 2024), but found that it resulted in decreased accuracy in our experiments. Therefore, we did not use it in the final experimental setup.

## C.3 DATASETS

Table 5 provides a detailed breakdown of the datasets used for training in every stage (stage I: *projector-training*, stage II: *warm-up*, stage III: *finetuning*). We maintain the same training pipeline and stages for all baselines and our Dirichlet-Prior Shaped models.

## C.4 HYPERPARAMETERS, IMPLEMENTATION, AND TRAINING DETAILS

This appendix outlines the hyperparameters, implementation specifics, and training procedures employed for the experiments discussed in Section 3.2.

All models were trained on a distributed setup utilizing either 4 or 8 NVIDIA A100 GPUs. A consistent total batch size of 128 was maintained across all experiments. When using 4 GPUs, the per-device batch size was set to 8, complemented by 4 gradient accumulation steps. In the 8-GPU

Table 5: Datasets used in training stages. On the first stage, we are training the adapter network. On the second stage, we train the whole network on a larger dataset including 30% of the data used for the Stage III.

| | Datasets | Size |
|---|---|---|
| **Stage I** | LLaVA 1.5-558k (Liu et al., 2024b) | 558k |
| **Stage II** | LLaVA 1.5-mix-665 (30%) (Liu et al., 2024b) SAM (30%) (Kirillov et al., 2023) Wikiart (30%) (Saleh & Elgammal, 2015) LVIS (Wang et al., 2023) ALLaVA (Chen et al., 2024) TextVQA (Singh et al., 2019) | 1,206k |
| **Stage III** | LLaVA 1.5-mix-665k (Liu et al., 2024b) SAM (Kirillov et al., 2023) Wikiart (Saleh & Elgammal, 2015) | 750k |

configuration, the per-device batch size remained 8, but with 2 gradient accumulation steps. For efficient distributed training, we leveraged DeepSpeed with ZeRO-2 offloading.

The models were optimized using the AdamW optimizer, configured with $\beta_1 = 0.9$ and $\beta_2 = 0.999$. The learning rate was varied across training stages: set to $1 \times 10^{-3}$ during Stage I, and reduced to $2 \times 10^{-5}$ for both Stage II and Stage III in all experiments. A cosine learning rate scheduler was used, with a warmup ratio of 0.03.

For our proposed method, the coefficient for the Dirichlet-Prior Shaping Loss (DPSL) was set to $\lambda = 0.01$. The baseline methods were implemented following the descriptions provided in their respective original publications, and we generally adopted the hyperparameters recommended by their authors. Specifically, the weight for the standard load-balancing loss (Shazeer et al., 2017; Fedus et al., 2022) was set to 0.01, and the weight for the z-loss (Zoph et al., 2022) regularizer was 0.001. Following the DeepSeek-V3 Technical Report (Wang et al., 2024; Liu et al., 2024a), we evaluated two update rates ($u = 0.001$ and $u = 0.0001$) for the auxiliary-loss-free DeepSeek strategy and selected the one that yielded the highest final accuracy, even though it produced less balanced routing than all other baselines as can be seen in Table 12. For the Drop-Upcycling (Nakamura et al., 2025) baseline, we encountered instabilities and training freezes with the initially recommended 50% drop rate settings. Consequently, we adjusted the ratios of re-initialized parameters. We found that a re-initialization ratio of 0.5 worked best for the 4-expert setup, but this value led to instabilities in the granular setup with 16-experts. Thus, for the 16-expert setup, we used a smaller re-initialization ratio of 0.2 to ensure stable training, and reported highest accuracies.

### C.5 COMPUTATIONAL OVERHEAD OF DPSL

Computing the DPSL loss introduces an additional minimal overhead during training. However, several factors help mitigate this overhead. First, the loss is computed over all tokens in a mini-batch (effective batch size $B = S \times T$), which is a highly parallelizable operation. Second, the gradient computation is efficient, as the derivative of the CDF used in the loss calculation is simply the PDF, which is already available from the forward pass. Third, empirical observations indicate that the router distributions converge to the target shape early in training and remain stable thereafter. Consequently, DPSL loss was applied only during the warm-up phase and relaxed during final fine-tuning, thereby minimizing its impact on overall training time.

To quantify this overhead precisely, we profiled the forward pass on a Qwen2.5-0.5B MoE model across varying expert counts (4, 8, and 16) using a Pytorch implementation. As shown in Table 6, the standard implementation introduces a modest overhead of 7–12% depending on the number of experts, primarily due to CPU-GPU transfers for SciPy-based Beta CDF computation. However, when using DeepSpeed with ZeRO-3 offloading (Table 7), this overhead drops significantly to just 1–5%. For larger models, such as Qwen2.5-1.5B, the relative cost becomes smaller as the constant-time CDF operation is dwarfed by the model's forward/backward pass. We anticipate this cost

will significancy be reduced with the upcoming native GPU support for Beta CDFs in PyTorch (`torch.special.betainc`), eliminating the need for device transfers.

Finally, DPSL is active only during the warm-up phase (Stage II), which accounts for ≈40% of our training pipeline. Using DeepSpeed, a 1-5% overhead in this stage translates to a trivial 0.4-2% increase in total wall-clock time.

Table 6: Qwen2.5-0.5B MoE forward pass overhead (avg. of 5 runs)

| Setting | w/o DPSL (ms) | w/ DPSL (ms) | Overhead |
|---|---|---|---|
| 4 experts | 189 | 203 | 14ms (+7%) |
| 8 experts | 272 | 299 | 27ms (+10%) |
| 16 experts | 434 | 487 | 52ms (+12%) |

Table 7: Qwen2.5-0.5B MoE forward pass overhead using DeepSpeed ZeRO-3 (avg. of 5 runs)

| Setting | w/o DPSL (ms) | w/ DPSL (ms) | Overhead |
|---|---|---|---|
| 4 experts | 502 | 507 | 5ms (+1%) |
| 8 experts | 797 | 825 | 28ms (+3.5%) |
| 16 experts | 1358 | 1436 | 78ms (+5%) |

### C.6 IMPLEMENTATION DETAILS FOR TASK-SPECIFIC EXPERT SPECIALIZATION

This appendix details the implementation for task-specific expert specialization, as referenced in Section 3.4. For this setup, using a 4-expert MoE model with top-2 routing, we partitioned the data utilized during the *warm-up* stage (Stage II) into four specific subsets: 1) data comprising text-only and image captions; 2) data focused on general question answering tasks; 3) data related to grounding tasks; and 4) a combined subset for OCR, chart understanding, and science-related tasks.

It is important to highlight a potential limitation inherent in such manual data partitioning, especially for vision-language modeling. The process of creating distinct, meaningful subsets is non-trivial and can inadvertently over-constrain the experts. For instance, many real-world vision-language tasks may benefit from, or even necessitate, knowledge derived from a combination of these defined categories (e.g., a chart-based question answering task might require OCR, chart understanding, and general QA capabilities). Consequently, this manual separation may restrict experts from learning broader, more synergistic representations, potentially leading to the sub-optimal performance observed in Table 2.

## D ABLATION STUDIES

### D.1 CONCENTRATION PARAMETER

For this ablation, we utilized the upcycled Llama3.2-1B model with 4 experts and top-2 routing. We performed a study over $\alpha \in \{0.75, 1.0, 1.25, 1.5\}$. The results in Table 8 show optimal performance at $\alpha = 0.75$, suggesting a benefit from priors encouraging slightly sparser routing than uniform. Ablation results for Qwen2-1.5B are shown in Table 9. We can observe that larger models are robust across a wider range of $\alpha$ values.

Table 8: The impact of the Dirichlet prior parameter $\alpha$ on Llama3.2-1B (2in4) performance.

| Prior | TextVQA | GQA | MM-Vet | MME-P | MME-C | VizWiz | MMB | Avg |
|---|---|---|---|---|---|---|---|---|
| $\alpha = 0.75$ | **52.82** | **60.98** | **31.7** | **1334.78** | 253.21 | **62.78** | **42.19** | **35.92** |
| $\alpha = 1.0$ | 51.68 | 60.84 | 30.2 | 1310.57 | 243.93 | 61.94 | 38.43 | 34.86 |
| $\alpha = 1.25$ | 51.48 | 60.53 | 29.2 | 1236.36 | 227.86 | 61.32 | 41.01 | 34.92 |
| $\alpha = 1.5$ | 51.45 | 60.85 | 31.4 | 1294.43 | **256.43** | 61.94 | 37.75 | 34.91 |

Table 9: The impact of the Dirichlet prior parameter $\alpha$ on Qwen2-1.5B (8in16) performance.

| Prior | TextVQA | GQA | MM-Vet | MME-P | MME-C | VizWiz | MMB | Avg |
|---|---|---|---|---|---|---|---|---|
| $\alpha = 0.75$ | 54.11 | **62.08** | 32.6 | 1427.13 | **280** | 66.20 | 43.18 | 37.03 |
| $\alpha = 1.0$ | **54.32** | 61.86 | 34.0 | 1421.90 | 265 | 65.98 | **43.54** | 37.25 |
| $\alpha = 1.25$ | 53.79 | 62.05 | 35.1 | **1428.87** | 276 | **66.48** | 41.53 | 37.14 |
| $\alpha = 1.5$ | 53.42 | 62.02 | **36.5** | 1402.43 | 270 | 65.70 | 42.31 | **37.28** |

## D.2 CONCENTRATION PARAMETER IN THE SPECIALIZATION SETTING.

We have additionally ablated the impact of $\alpha$ concentration parameter on the model performance in the modality specialized setting. We have considered four different setups: *Setting I* encouraging two experts for vision and two for language with concentration values $\alpha_{lm} = (0.75, 0.75, 1.25, 1.25)$ and $\alpha_v = (1.25, 1.25, 0.75, 0.75)$, *Setting II* – similar to the previous setup but with different $\alpha$ values $\alpha_{lm} = (1.0, 1.0, 1.25, 1.25)$ and $\alpha_v = (1.25, 1.25, 1.0, 1.0)$, *Setting III* encouraging three experts to specialize in vision and one in language with $\alpha_{lm} = (0.75, 0.75, 0.75, 1.25)$ and $\alpha_v = (1.25, 1.25, 1.25, 0.75)$, and, finally, *Setting IV* encouraging one expert to specialize in vision and thee in language with $\alpha_{lm} = (0.75, 1.25, 1.25, 1.25)$ and $\alpha_v = (1.25, 0.75, 0.75, 0.75)$.

Table 10: The impact of the Dirichlet prior parameter $\alpha$ on Llama3.2-1B (2in4) performance in the modality specialized setting.

| Setting | TextVQA | GQA | MME-P | MME-C | VizWiz | MMB | Avg |
|---|---|---|---|---|---|---|---|
| I | 51.8 | 61.4 | 1305 | 285 | 64.0 | 42.9 | 36.85 |
| II | 51.7 | 61.2 | 1314 | 261 | 63.9 | 42.1 | 36.65 |
| III | 51.4 | 60.7 | 1310 | 269 | 63.1 | 39.4 | 35.94 |
| IV | 50.7 | 56.0 | 1240 | 261 | 62.3 | 40.2 | 35.02 |

The results summarized in Table 10 suggest that DPSL is robust to minor changes of the $\alpha$ values, as long as the fundamental architectural prior is preserved. However, ill-conceived priors that encourage unbalancing the expert allocation lead to a degraded model performance (Settings III and IV). Interestingly, the performance drop is not symmetric. Starving the model of vision experts (Setting IV) is significantly more detrimental than starving it of language experts (Setting III). This is intuitive, as VLM inputs typically consist of a large number of vision tokens sourced from the image and relatively few language tokens obtained from the question. Restricting the model's capacity to process the larger modality creates a more severe bottleneck.

## D.3 IMPACT OF REGULARIZATION WEIGHT.

We have studied the effect of $\lambda$ loss regularization weight. The results summarized in Table 11 below suggest that the value $\lambda = 0.01$ yields the best performance. Based on this ablation study, we fixed this value for all subsequent experiments.

Table 11: The impact of the loss regularization weight parameter $\lambda$ on Llama3.2-1B (2in4) performance.

| $\lambda$ value | TextVQA | GQA | MM-Vet | MME-P | MME-C | VizWiz | MMB | Avg |
|---|---|---|---|---|---|---|---|---|
| $\lambda = 0.001$ | 51.3 | 60.9 | 28.7 | 1286 | 236.8 | 60.4 | **44.0** | 35.2 |
| $\lambda = 0.01$ | **52.8** | **61.0** | **31.7** | **1335** | **253.2** | **62.8** | 42.2 | **35.9** |
| $\lambda = 0.1$ | 50.9 | 60.6 | 30.5 | 1261 | 241.4 | 61.2 | 39.6 | 34.8 |

## D.4 Visualization of Routing Score Distributions

As discussed in Section 3.6, we analyze the impact of different upcycling and regularization strategies on the routing score distributions within our upcycled VLM MoEs. Figure 7 provides a visualization of the routing score distributions for 4 experts at the 12th intermediate layer of a Llama3.2-1B model configured with 4 experts and top-2 routing. The routing scores presented in this visualization were collected during model evaluation on the MM-Vet benchmark (Yu et al., 2024).

The figure compares several training approaches: Standard sparse upcycling (Komatsuzaki et al., 2023), load-balancing (Shazeer et al., 2017; Fedus et al., 2022), auxiliary-loss-free DeepSeek balancing (Wang et al., 2024; Liu et al., 2024a), and z-loss (Zoph et al., 2022). Notably, these approaches tend to produce similar routing score distributions across the experts. Each distribution exhibits a prominent peak around a score of 0.25, corresponding to a uniform probability distribution if the router were to assign equal preference to each of the four available experts. This suggests a lack of strong differentiation or specialization among them.

In contrast, our DPSL, when applied with a symmetric prior where $\alpha_k = 1.5$ for all experts, results in visibly different routing score distributions. The distributions generated by DPSL are more dispersed and cover a wider range of score values. This indicates that DPSL encourages the router to make more varied and potentially more confident assignments, fostering a greater degree of specialization or differentiation in how tokens are directed to the experts.

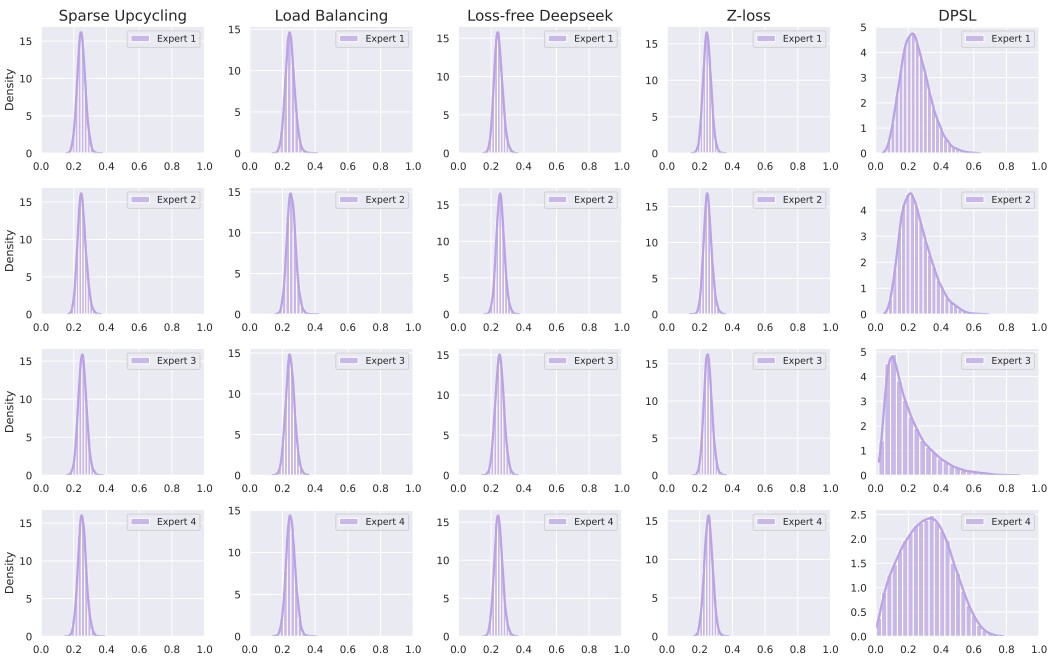

Figure 7: Routing score distributions at layer 12 of an upcycled Llama3.2-1B model (4-expert, top-2 routing). Each column represents a different upcycling/regularization method, and each row displays the distribution for one of the four experts under that method.

## D.5 Expert Utilization Analysis

To analyze the expert utilization load, we measure Coefficient of Variation (CoV). As the results show, DPSL achieves low CoV scores, competitive with explicit load-balancing techniques. Although DPSL does not contain an explicit load-balancing term, a symmetric Dirichlet prior intrinsically encourages a balanced load distribution, which we observe across layers.

In all of our experiments, DPSL successfully prevented expert collapse and significant utilization imbalance. We note, however, that none of the baseline methods exhibited severe imbalance issues in our experiments.

Table 12: Coefficient of Variation of expert loads for Llama3.2-1B (2in4) model. Lower values indicate more balanced utilization.

| Layer | Sparse Upcycling | Load Balancing | z-loss | DeepSeek | DPSL (Ours) |
|---|---|---|---|---|---|
| Layer 4 | 0.071 | 0.070 | 0.088 | 0.440 | 0.035 |
| Layer 8 | 0.075 | 0.126 | 0.064 | 0.191 | 0.069 |
| Layer 12 | 0.072 | 0.091 | 0.031 | 0.397 | 0.057 |
| Layer 16 | 0.071 | 0.053 | 0.103 | 0.229 | 0.110 |

## D.6 EXPERT SPECIALIZATION PATTERNS

We analyze expert activation similarity patterns for Llama3.2-1B (2in4) across layers 3, 6, 9, and 12 using cosine similarity between expert outputs on 50 MMVet randomly selected sequences. Low similarity values ($< 0.4$, red) indicates distinct expert specialization, while higher similarity values ($0.4 - 0.8$, yellow-green) suggest overlapping expert behaviors and reduced differentiation. As shown in Figure 8, DPSL demonstrates superior expert differentiation (average similarity: 0.39). In contrast, Load Balancing loss and Router z-loss shows progressive expert convergence from early to deep layers (average similarity: 0.57 and 0.59, respectively). These results clearly indicate that our method effectively prevents expert redundancy and maintains expert specialization across all layers where traditional auxiliary losses struggle to maintain diversity.

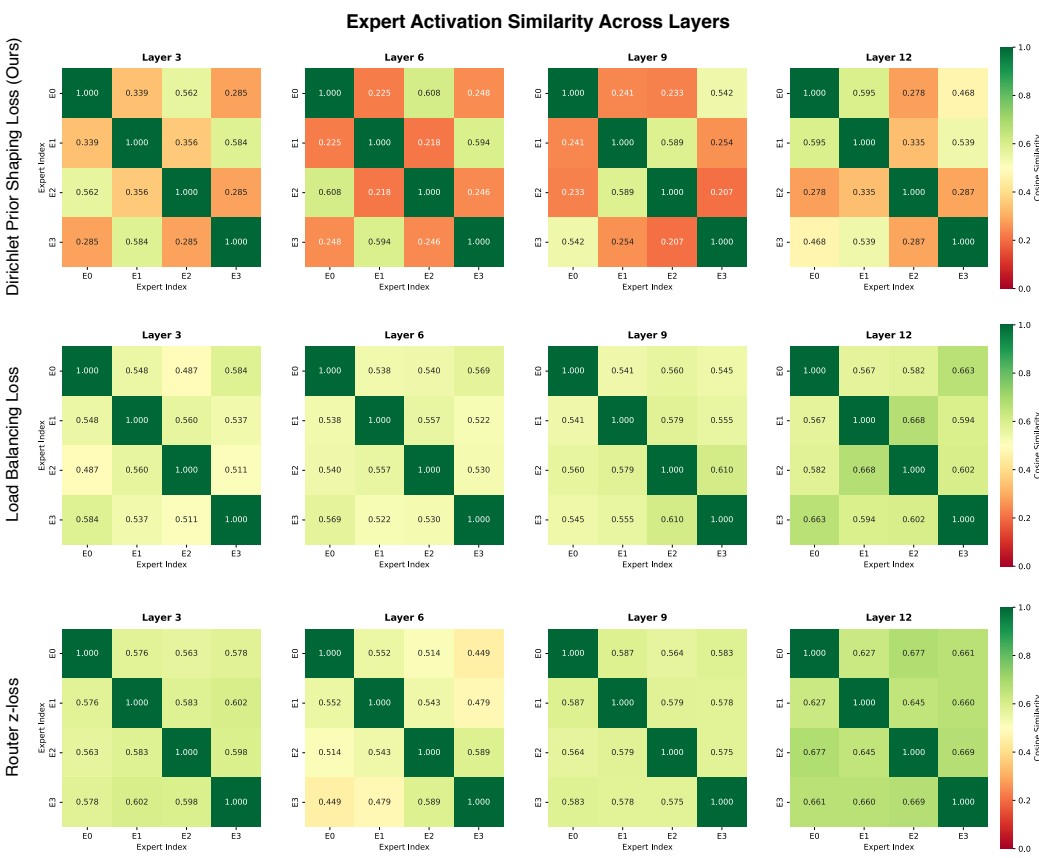

Figure 8: Expert activation similarity scores across layers 3, 6, 9, and 12. Our DPSL (top row) achieves the lowest average similarity (0.39) for these layers compared to Load Balancing loss (0.57) and Router z-loss (0.59), demonstrating superior expert specialization.

