# OpenReview forum: "Dirichlet-Prior Shaping: Guiding Expert Specialization in Upcycled MoEs"
_ICLR.cc/2026/Conference — Submitted to ICLR 2026_

### Official Review · Reviewer_ZHna · 2025-10-29

**Soundness:** 3
**Presentation:** 2
**Contribution:** 2
**Rating:** 4
**Confidence:** 2

**Summary:**

This paper proposes a new regularization loss for Upcycled MoE training to address the issue of uniform routing weights. Specifically, the DPSL regularizes the distance between the empirical CDF and a target Beta CDF for each expert category.

**Strengths:**

1. The motivation is clear, and I believe that the distribution of routing probabilities is a critical issue in upcycled MoE models.
2. The proposed method aligns well with this motivation, and performance improvements are observed in several scenarios.

**Weaknesses:**

1. It is unclear why shaping the routing probability distribution to follow a Dirichlet distribution is the optimal choice. Could the authors provide stronger theoretical or empirical justification for this design decision? Additionally, the reported performance gains appear marginal in most cases, which raises questions about the practical significance of the proposed approach.

2. According to Table 8, DPSL achieves better load balancing than the standard load-balancing loss. Could the authors clarify the underlying reason for this improvement? Furthermore, would this advantage persist in Modality-Specific and Task-Specific settings? For a more comprehensive evaluation, it would also be helpful to report training and inference times.

3. While the hyperparameter $\alpha$ has a substantial impact on the logit distribution, Tables 4 and 5 suggest that identifying an optimal setting is non-trivial. This may limit the practical applicability of the method in real-world scenarios where extensive tuning is often infeasible.

**Questions:**

See weakness

---

> ### Author Response · Authors · 2025-11-22
> **Response to the Reviewer ZHna (1/2)**
>
> We thank the reviewer for their constructive criticism and for recognizing the importance of router probability distribution in upcycled MoE models. We appreciate that the reviewer found our method well-aligned with the problem motivation and noted the performance improvements. Below, we address the specific questions and concerns raised.
>
> **W1)** We clarify that the choice of the Dirichlet distribution is not arbitrary but mathematically foundational.
>
> **Theoretical Justification:** The Dirichlet distribution is the **conjugate prior for categorical distributions**. It is the mathematically natural distribution for representing beliefs about categorical probability vectors. In Bayesian statistics, it is the unique, canonical choice for modeling probability vectors over the simplex. Its concentration parameters $\alpha$ provide explicit, interpretable control over the distribution's shape, allowing us to smoothly interpolate between uniform (balanced) and sparse (specialized) behaviors within a single unified framework. This is why Dirichlet priors are the standard in mixture models (Antoniak, 1974) [1] and topic modeling (LDA; Blei et al., 2003) [2,3]. There is effectively no other distribution with comparable theoretical grounding and practical flexibility for this specific purpose.
>
> We have updated the manuscript (`Section 2.1`) to explicitly state the conjugate prior relationship to prevent future ambiguity.
>
> [1] Antoniak, Charles E. "Mixtures of Dirichlet processes with applications to Bayesian nonparametric problems." The annals of statistics (1974)
>
> [2] Blei, David M., Andrew Y. Ng, and Michael I. Jordan. "Latent dirichlet allocation." JMLR (2003)
>
> [3] Wallach, Hanna, David Mimno, and Andrew McCallum. "Rethinking LDA: Why priors matter." NeurIPS (2009)
>
> **Empirical Justification:** The paper provides extensive evidence that this theoretical choice translates to practical gains:
>
> 1.  DPSL achieves the best average performance across *all* tested backbones (Qwen2, Phi-3, Llama3.2) and configurations (`Table 2`).
> 2.  Ablation (`Table 8`) show stable performance across a wide range of $\alpha$ values.
> 3.  Our Coefficient of Variation (CoV) analysis (`Table 12`) confirms that the Dirichlet prior naturally induces balanced loads without auxiliary losses.
>
> **New Empirical Results:**
>
>   4.  New experiments in unsupervised clustering (`Section 2.3`) show gains of 4.6–14.9\%, proving the prior's effectiveness beyond MoE routing.
>   5.  New expert activation similarity analysis (`Section 3.6 & Appendix D.6`) proving distinct expert specialization.
>
> Please see our Global response and related sections in the revised manuscript for more information on these new experiments.
>
>
>
> **W2)** We thank the reviewer for raising this point. We believe the improvements are practically meaningful for two reasons.
>
> First, DPSL is the only method that yields the best average performance across *all* three backbones (Qwen2, Phi-3, Llama3.2) and both MoE configurations (2in4 and 8in16). In contrast, the "runner-up" varies significantly by setting: e.g., a method that is competitive on Phi-3 2in4 can lag behind DPSL by around 1–1.5 points on Qwen2 models. This suggests that DPSL is not tuned to a single favorable configuration but serves as a robust regularizer across architectures and granularities.
>
> Second, for vision-language models at this scale, consistent improvements of up to 1\% are substantial. Training pipelines for modern VLMs (like LLaVA) are already heavily optimized, making further gains challenging.
>
> **W3)** We thank the reviewer for highlighting this result. We view this intrinsic load balancing as a key strength of our method rather than a weakness.
>
> With a symmetric prior, DPSL does not only encourage confident, well‑separated router outputs, but also implicitly penalizes systematic over‑use or under‑use of experts at the batch level, because the empirical routing histogram is matched to a symmetric Dirichlet target.
>
> In Modality‑Specific and Task‑Specific settings, the notion of “balanced” load becomes conditional on the data distribution: asymmetric priors intentionally bias routing so that some experts are used more often for certain token types or tasks. As a result, overall CoV across all tokens can increase by design, but this is not a weakness, it reflects the desired specialization pattern. In practical scenarios such as on‑device deployment, such structured imbalance can be more beneficial: e.g., biasing a subset of experts toward coding queries allows caching exactly those experts in fast memory, while rarely used experts remain on slower storage.

---

> > ### Author Response · Authors · 2025-11-22
> > **Response to the Reviewer ZHna (2/2)**
> >
> > **W4)** We appreciate the reviewer’s inquiry into training and inference costs. We have included a detailed analysis of the computational costs in `Appendix C.5` of the revised manuscript, which we summarize below. Please refer to `Table 6` (standard training) and `Table 7` (DeepSpeed Zero-3) in that section for the specific profiling data.
> >
> > The primary cost in our initial code was the CPU-GPU transfer required to compute Beta CDFs via NumPy/SciPy. For this rebuttal, we optimized the implementation to minimize these transfers and redundant reshaping. As shown in `Table 6` and `7` in `Appendix C.5`, the computational cost of DPSL on a Qwen2.5-0.5B model is as follows:
> > *   **Standard Training:** 7-12\% overhead depending on expert count.
> > *   **DeepSpeed Zero-3:** Overhead drops to 1-5\%.
> >
> > For larger models (Qwen2.5-1.5B), the relative gap shrinks further to negligible levels, as the CDF computation cost is constant while the model's forward/backward pass cost grows.
> >
> > DPSL is active only during the warm-up phase (Stage II), which accounts for ~40\% of our training pipeline. Using DeepSpeed, a 1-5\% overhead in this stage translates to a trivial **0.4-2\% increase in total wall-clock time**. There is **zero** cost at inference time.
> >
> > **W5)** While DPSL introduces $\alpha$ (Dirichlet concentration) and $\lambda$ (regularization weight), the tuning burden is comparable to existing MoE regularization methods, which also require hyperparameter selection (e.g., load-balancing loss coefficients). In our setup, $\alpha$ plays the role of an interpretable prior rather than a free tuning knob: without specifying a prior, there would be no way to encode any desired routing behavior. To guide practitioners, we have added a new subsection **"Practical guidelines for selecting $\alpha$"** to `Section 3.5` in the revised manuscript, detailing our recommended strategy.
> >
> > In practice, we recommend $\alpha=(1,1,..., 1)$ as a universal starting point across model sizes and architectures. In our experiments, we used $\alpha=1$ for Qwen2 and Phi-3, while observing that the smaller Llama3.2-1B benefits from slightly stronger specialization ($\alpha=0.75$). We observe that this extends beyond MoEs. In our new unsupervised clustering experiments, we found that moderate priors around $\alpha = 1$ consistently work well, and even deliberately misspecified priors yield gains over baselines. This confirms that DPSL provides beneficial inductive bias without requiring fine-grained tuning.
> >
> > **Summary of Practical Guidelines (`Section 3.5`):**
> > 1.  **Default:** Use symmetric $\alpha=1$ for all experts to achieve balanced specialization (our default across most models).
> > 2.  **Small Models:** Optionally reduce to $\alpha=0.75$ for smaller models (<2B parameters) to encourage stronger specialization. Larger models are stable across $\alpha \in [0.75, 1.5]$.
> > 3.  **Asymmetric Needs:** For specific preferences (e.g., modality imbalance), start from the base value and add $\approx 0.5$ to the preferred component (e.g., $\alpha = (1.5, 1, 1)$) to softly bias probability mass.
> > 4.  **Weighting:** We find $\lambda = 0.01$ to be a robust default for the loss weight across all settings, with tuning effort comparable to standard MoE load-balancing coefficients.
> >
> >
> > Please let us know if these clarifications and the additional results in the revised paper adequately address your concerns. We are happy to provide any further clarification.

---

### Official Review · Reviewer_Tjkt · 2025-10-30

**Soundness:** 3
**Presentation:** 3
**Contribution:** 3
**Rating:** 6
**Confidence:** 3

**Summary:**

This paper addresses the challenge of poor expert specialization in Mixture-of-Experts (MoE) models created by upcycling pre-trained dense models. The authors argue that naive weight replication leads to homogeneous experts and low-confidence routing, which conventional regularization methods like load-balancing or z-loss fail to adequately resolve. To this end, they introduce Dirichlet-Prior Shaping Loss (DPSL), a novel regularization technique that directly shapes the router's output probability distribution. DPSL matches the empirical cumulative distribution function (CDF) of expert assignment probabilities to a target Beta distribution, which is the marginal of a chosen Dirichlet prior. This approach allows for fine-grained control over expert utilization and specialization. The authors demonstrate that symmetric priors can enforce confident and balanced routing, while asymmetric priors can be used to instill inductive biases, such as encouraging experts to specialize in specific modalities (e.g., vision vs. language) in Vision-Language Models (VLMs). Through extensive experiments on upcycled VLM MoEs with various backbones (Qwen2, Phi3, Llama3.2), the paper shows that DPSL consistently outperforms existing upcycling and regularization techniques on a suite of standard vision-language benchmarks.

**Strengths:**

- DPSL is a clean approach that provides fine-grained control over router behavior. It moves beyond simple heuristics like load balancing and allows practitioners to instill complex and desirable statistical properties into the routing mechanism.
- The authors have conducted an extensive set of experiments across three different modern LLM backbones, two MoE configurations, and six benchmarks.
- The analysis of router output distributions (e.g., Figure 3 and Appendix C.5) provides valuable qualitative insight into why DPSL works better than alternatives, by encouraging more confident and diverse routing assignments.

**Weaknesses:**

- The paper focuses exclusively on upcycled VLMs. While the authors claim DPSL is a general tool, they provide no evidence of its efficacy for training MoEs from scratch or in other domains like language-only models. Demonstrating its utility beyond the upcycling VLM setting would significantly strengthen the claims of generality.

- The paper reports a 10-15% computational overhead during training, which is not negligible. The mitigation strategy of only applying it during a warm-up phase is reasonable, but a more detailed analysis of the impact on total training time would be welcome. Furthermore, DPSL introduces new hyperparameters (α and λ), and the ablation study shows that different model backbones prefer different settings (e.g., α=0.75 for Llama3.2 vs. α=1.0 for Qwen2). This suggests that some degree of architecture-specific tuning may be required, which could be a practical drawback.

**Questions:**

- The lack of multiple seeds is a concern. Could you provide results for at least one model configuration (e.g., Llama3.2-1B 2in4) across 3-4 seeds for your method and the top-performing baselines (e.g., DeepSeek balancing, Drop-Upcycling)?
- The results in Table 2 show that explicitly guiding specialization toward pre-defined task subsets (both with BTX and DPSL) underperforms the general-purpose symmetric DPSL. The paper speculates this is due to "nonoptimal data subsets." An alternative hypothesis is that enforcing hard, fine-grained specialization is simply the wrong inductive bias for these multi-faceted VLM tasks. Could you comment on this alternative interpretation? Does this result suggest a potential limitation of highly specialized expert priors?
- Could you discuss the expected behavior and potential challenges of applying DPSL to training MoEs from scratch? In that setting, the router and experts co-evolve from a random initialization. Would you expect DPSL to be more or less effective compared to its application in the upcycling setting where the router's main job is to break symmetry?

---

> ### Author Response · Authors · 2025-11-22
> **Response to the Reviewer Tjkt (1/3)**
>
> We thank the reviewer for the thoughtful feedback. We are pleased that the reviewer found DPSL to be a clean approach for fine-grained router control and appreciated our extensive experiments and the valuable qualitative insights from our router output analysis. Below, we address the specific questions and concerns raised.
>
> **W1)** We thank the reviewer for this constructive feedback. We agree that demonstrating generality is crucial. Since Reviewers **6fqQ** and **2RyL** similarly requested evidence of applicability beyond MoE routing, we prioritized applying DPSL to a fundamentally different domain involving categorical distributions rather than just another MoE variant. This provides a stricter test of whether the method is a general-purpose tool. To this end, we applied DPSL to **unsupervised deep clustering**, a canonical task completely distinct from MoE architectures.
>
> We integrated DPSL into established clustering baselines, SwAV (Caron et al., 2020) and SeCu (Yan et al., 2023), using synthetic datasets with imbalanced cluster sizes (ratios 4:1:1, 5:2:1, and 5:3:1). While standard methods often favor balanced partitions, real-world clusters are frequently asymmetric. We used DPSL to impose *moderate asymmetric Dirichlet priors* on the cluster assignments.
>
> **Results:** As detailed in the new `Section 2.3`, DPSL yielded substantial accuracy gains across all regimes for both SwAV ($4.6–14.9$\%) and SeCu ($5.9–12.6$\%), as shown below (mean $\pm$ standard error across three random seeds):
>
> | Setting | SwAV | SwAV+DPSL | SeCu | SeCu+DPSL |
> | :--- | :--- | :--- | :--- | :--- |
> | Non-overlapping | 84.4 ± 0.8\% | **99.4 ± 0.1\%** (+14.9\%) | 88.4 ± 11.4\% | **94.3 ± 5.5\%** (+5.9\%) |
> | Overlapping | 86.7 ± 1.1\% | **94.1 ± 0.2\%** (+7.4\%) | 70.7 ± 3.8\% | **83.3 ± 1.7\%** (+12.6\%) |
> | Elongated | 87.7 ± 0.7\% | **92.3 ± 0.5\%** (+4.6\%) | 76.8 ± 7.1\% | **87.7 ± 3.9\%** (+10.8\%) |
>
> **Robustness to Prior Misspecification:** To test robustness, we also used a fixed, sub-optimal prior ($\alpha = (1.5, 1, 1)$) across all settings. As shown in `Appendix B` `Table 4`, DPSL still consistently improved performance over the baselines, confirming it serves as a beneficial structural prior even when the target is only approximately known.
>
> Full experimental details (`section 2.3` and `Appendix B`), visualizations of the qualitative impact on cluster shapes (`Figure 5`), and the rationale for prior selection are provided in the revised manuscript. These results validate DPSL as a general-purpose regularizer for shaping categorical distributions.
>
>
>
> **W2)** We appreciate the reviewer’s detailed inquiry into these costs. We have included a detailed analysis of the computational costs in `Appendix C.5` of the revised manuscript, which we summarize below. Please refer to `Table 6` (standard training) and `Table 7` (DeepSpeed Zero-3) in that section for the specific profiling data. Through an optimized implementation, we managed to reduce the overall increase in **total wall-clock time to a trivial $0.4-2$\%**.
>
> **Optimized Implementation \& Profiling:** The primary cost in our initial code was the CPU-GPU transfer required to compute Beta CDFs via NumPy/SciPy. For this rebuttal, we optimized the implementation to minimize these transfers and redundant reshaping. As shown in `Table 6` and `7` in `Appendix C.5`, the computational cost of DPSL on a Qwen2.5-0.5B model is as follows:
> *   **Standard Training:** 7-12\% overhead depending on expert count.
> *   **DeepSpeed Zero-3:** Overhead drops to 1-5\%.
>
> For larger models (e.g. Qwen2.5-1.5B), the relative gap shrinks further to negligible levels, as the CDF computation cost is constant while the model's forward/backward pass cost grows.
>
>
>
> **Vanishing Cost with Native GPU Support:** Crucially, the current overhead is an artifact of using SciPy for CDFs. PyTorch is actively adding native support for special functions (e.g., `torch.special.betainc`), and other frameworks like JAX/TensorFlow already support this on GPU. Once fully native, the overhead will effectively vanish, as the operation is faster and avoids CPU-GPU transfer overhead.
>
> **End-to-End Impact:**
> DPSL is active only during the warm-up phase (Stage II), which accounts for ~40\% of our training pipeline. Using DeepSpeed, a 1-5\% overhead in this stage translates to a trivial **0.4-2\% increase in total wall-clock time**. There is *zero* cost at inference time.

---

> > ### Author Response · Authors · 2025-11-22
> > **Response to the Reviewer Tjkt (2/3)**
> >
> > **W3)** While DPSL introduces $\alpha$ (Dirichlet concentration) and $\lambda$ (regularization weight), the tuning burden is comparable to existing MoE regularization methods, which also require hyperparameter selection (e.g., load-balancing loss coefficients). In our setup, $\alpha$ plays the role of an interpretable prior rather than a free tuning knob: without specifying a prior, there would be no way to encode any desired routing behavior. To guide practitioners, we have added a new subsection **"Practical guidelines for selecting $\alpha$"** to `Section 3.5` in the revised manuscript, detailing our recommended strategy.
> >
> > In practice, we recommend $\alpha=(1,1,..., 1)$ as a universal starting point across model sizes and architectures. In our experiments, we used $\alpha=1$ for Qwen2 and Phi-3, while observing that the smaller Llama3.2-1B benefits from slightly stronger specialization ($\alpha=0.75$). We observe that this extends beyond MoEs. In our new unsupervised clustering experiments, we found that moderate priors around $\alpha = 1$ consistently work well, and even deliberately misspecified priors yield gains over baselines. This confirms that DPSL provides beneficial inductive bias without requiring fine-grained tuning.
> >
> > **Summary of Practical Guidelines (`Section 3.5`):**
> >
> > 1.  **Default:** Use symmetric $\alpha=1$ for all experts to achieve balanced specialization (our default across most models).
> > 2.  **Small Models:** Optionally reduce to $\alpha=0.75$ for smaller models (<2B parameters) to encourage stronger specialization. Larger models are stable across $\alpha \in [0.75, 1.5]$.
> > 3.  **Asymmetric Needs:** For specific preferences (e.g., modality imbalance), start from the base value and add $\approx 0.5$ to the preferred component (e.g., $\alpha = (1.5, 1, 1)$) to softly bias probability mass.
> > 4.  **Weighting:** We find $\lambda = 0.01$ to be a robust default for the loss weight across all settings, with tuning effort comparable to standard MoE load-balancing coefficients.
> >
> >
> > **Q1)** We agree that reporting variance across seeds strengthens empirical evidence. However, as detailed in `Appendix C.4`, each VLM-MoE run requires 8 GPUs for several days. A full multi-seed study across all backbones and granularities would consume weeks of compute, which is prohibitive for our computational budget.
> >
> > To address this, we commit to adding multi-seed results for a representative configuration (e.g., Llama3.2-1B 2in4: DPSL vs. top baseline) in the camera-ready version. We also note that our new unsupervised clustering experiments (`Section 2.3`) already report mean $\pm$ standard error over three seeds, demonstrating that DPSL consistently yields stable gains across random initializations.
> >
> >
> > **Q2)** We appreciate this alternative interpretation and agree that it is plausible. Manually partitioning multi-faceted VLM tasks (e.g., "OCR" vs "reasoning") imposes a hard inductive bias that may conflict with the reality that real-world queries often require a blend of capabilities. The underperformance of task-specific priors in `Table 3` likely reflects this misalignment, where rigid separation prevents necessary knowledge sharing. We added the reviewer's point regarding "multi-faceted nature of VLM tasks" explaining this reduced performance in the revised manuscript.
> >
> > At the same time, we view this as a limitation of the particular task partitions we used, not of DPSL itself. DPSL’s contribution is to provide a flexible mechanism for injecting priors when appropriate: the same framework supports symmetric (general-purpose), modality-specific, and task-specific priors.
> >
> > Finally, there are scenarios where explicit task priors can be beneficial even if they slightly reduce aggregate accuracy, for example in resource-constrained deployments. In a *4-in-32* MoE served on a device where only 16 experts fit in fast memory, a task-specific prior could bias a specific task “e.g. coding” queries toward a *16-expert* subset, effectively inducing *4-in-16* behavior for that task. This trades a small potential loss in average accuracy for a large gain in cache hit rate and throughput.

---

> > ### Author Response · Authors · 2025-11-22
> > **Response to the Reviewer Tjkt (2/3)**
> >
> > **W3)** While DPSL introduces $\alpha$ (Dirichlet concentration) and $\lambda$ (regularization weight), the tuning burden is comparable to existing MoE regularization methods, which also require hyperparameter selection (e.g., load-balancing loss coefficients). In our setup, $\alpha$ plays the role of an interpretable prior rather than a free tuning knob: without specifying a prior, there would be no way to encode any desired routing behavior. To guide practitioners, we have added a new subsection **"Practical guidelines for selecting $\alpha$"** to `Section 3.5` in the revised manuscript, detailing our recommended strategy.
> >
> > In practice, we recommend $\alpha=(1,1,..., 1)$ as a universal starting point across model sizes and architectures. In our experiments, we used $\alpha=1$ for Qwen2 and Phi-3, while observing that the smaller Llama3.2-1B benefits from slightly stronger specialization ($\alpha=0.75$). We observe that this extends beyond MoEs. In our new unsupervised clustering experiments, we found that moderate priors around $\alpha = 1$ consistently work well, and even deliberately misspecified priors yield gains over baselines. This confirms that DPSL provides beneficial inductive bias without requiring fine-grained tuning.
> >
> > **Summary of Practical Guidelines (`Section 3.5`):**
> >
> > 1.  **Default:** Use symmetric $\alpha=1$ for all experts to achieve balanced specialization (our default across most models).
> > 2.  **Small Models:** Optionally reduce to $\alpha=0.75$ for smaller models (<2B parameters) to encourage stronger specialization. Larger models are stable across $\alpha \in [0.75, 1.5]$.
> > 3.  **Asymmetric Needs:** For specific preferences (e.g., modality imbalance), start from the base value and add $\approx 0.5$ to the preferred component (e.g., $\alpha = (1.5, 1, 1)$) to softly bias probability mass.
> > 4.  **Weighting:** We find $\lambda = 0.01$ to be a robust default for the loss weight across all settings, with tuning effort comparable to standard MoE load-balancing coefficients.
> >
> >
> > **Q1)** We agree that reporting variance across seeds strengthens empirical evidence. However, as detailed in `Appendix C.4`, each VLM-MoE run requires 8 GPUs for several days. A full multi-seed study across all backbones and granularities would consume weeks of compute, which is prohibitive for our computational budget.
> >
> > To address this, we commit to adding multi-seed results for a representative configuration (e.g., Llama3.2-1B 2in4: DPSL vs. top baseline) in the camera-ready version. We also note that our new unsupervised clustering experiments (`Section 2.3`) already report mean $\pm$ standard error over three seeds, demonstrating that DPSL consistently yields stable gains across random initializations.
> >
> >
> > **Q2)** We appreciate this alternative interpretation and agree that it is plausible. Manually partitioning multi-faceted VLM tasks (e.g., "OCR" vs "reasoning") imposes a hard inductive bias that may conflict with the reality that real-world queries often require a blend of capabilities. The underperformance of task-specific priors in `Table 3` likely reflects this misalignment, where rigid separation prevents necessary knowledge sharing. We added the reviewer's point regarding "multi-faceted nature of VLM tasks" explaining this reduced performance in the revised manuscript.
> >
> > At the same time, we view this as a limitation of the particular task partitions we used, not of DPSL itself. DPSL’s contribution is to provide a flexible mechanism for injecting priors when appropriate: the same framework supports symmetric (general-purpose), modality-specific, and task-specific priors.
> >
> > Finally, there are scenarios where explicit task priors can be beneficial even if they slightly reduce aggregate accuracy, for example in resource-constrained deployments. In a *4-in-32* MoE served on a device where only 16 experts fit in fast memory, a task-specific prior could bias a specific task “e.g. coding” queries toward a *16-expert* subset, effectively inducing *4-in-16* behavior for that task. This trades a small potential loss in average accuracy for a large gain in cache hit rate and throughput.

---

> > > ### Author Response · Authors · 2025-11-22
> > > **Response to the Reviewer Tjkt (3/3)**
> > >
> > > **Q3)** This is an excellent question. When training MoEs from scratch, the primary challenge is avoiding early expert collapse when both router and experts are untrained. In this regime, enforcing balanced exploration is more critical than inducing immediate specialization.
> > >
> > > A symmetric Dirichlet prior ($\alpha = 1$) is ideal for this phase: it encourages uniform expert usage without the rigid constraints of auxiliary load-balancing losses. As training progresses and experts begin to differentiate, one could naturally transition to asymmetric priors to guide specialization, though a symmetric prior often suffices to maintain healthy routing diversity (as shown in our CoV analysis).
> > >
> > > While upcycling requires the router to *break* symmetry between identical experts, training from scratch requires the router to *maintain* diversity while experts co-evolve. We expect DPSL to be equally effective in this setting because it directly regularizes the routing distribution shape, preventing the "winner-take-all" collapse common in early MoE training. We have added this discussion to the revised manuscript to clarify DPSL’s broader applicability.
> > >
> > > Please let us know if these clarifications and the additional results in the revised paper adequately address your concerns. We are happy to provide any further clarification.

---

### Official Review · Reviewer_6fqQ · 2025-10-31

**Soundness:** 3
**Presentation:** 3
**Contribution:** 3
**Rating:** 6
**Confidence:** 3

**Summary:**

The article proposes a novel router regularization technique, Dirichlet-Prior Shaping Loss (DPSL), to address the issue of insufficient specialization among experts in sparse mixture-of-experts (MoE) models. The paper analyzes the routing problems of low confidence and weak discrimination in conventional upcycled MoEs. To improve this, the authors propose matching the routing probability distribution with the target Dirichlet prior, flexibly controlling the balance and specialization level of experts.

**Strengths:**

- The method allows flexible control over the trade-off between expert balance and specialization by aligning router outputs with a tunable Dirichlet prior.

- Experimental results demonstrate its strong performance.

- The presentation is clear and easy to understand.

**Weaknesses:**

- The DPSL method shows good performance on various benchmarks but relies on selecting the appropriate Dirichlet prior (e.g., α=0.75 being more effective for Llama3.2-1B). This dependence may increase the hyperparameter tuning effort during training. It is recommended to further analyze the parameter selection process to help readers understand the rationale behind the choices.

- The authors mention in the abstract that DPSL is also a general-purpose tool. It is recommended to further elaborate on the potential applications of DPSL in other areas in the main text to demonstrate its broad applicability.

**Questions:**

- Consistency of terminology, it is recommended to unify the use of either "asymmetric priors" or "non-symmetric priors" to ensure consistency and accuracy in terminology.

- In Table 1, there is an inaccuracy in the labeling of the second-best data; please correct it.

---

> ### Author Response · Authors · 2025-11-22
>
> We thank the reviewer for the thoughtful feedback and for accurately summarizing our work as a flexible method for controlling expert balance and specialization. We are encouraged by the feedback regarding our clear presentation and strong experimental results. Below, we address the constructive suggestions on hyperparameter selection and broad applicability.
>
> **W1)** We agree that clear guidance on selecting the Dirichlet concentration $\alpha$ is essential for practical adoption. To address this, we have added a new subsection **"Practical guidelines for selecting $\alpha$"** to `Section 3.5` in the revised manuscript, detailing our recommended strategy.
>
> We recommend $\alpha = (1, 1, ..., 1)$ as a universal starting point across model sizes and architectures. This symmetric prior corresponds to a uniform Dirichlet distribution, promoting balanced expert utilization with moderate specialization. In our experiments, we used $\alpha=1$ for Qwen2 and Phi-3, while observing that the smaller Llama3.2-1B benefits from slightly stronger specialization ($\alpha=0.75$). We emphasize that this is a single scalar hyperparameter per backbone, and our ablations (`Tables 8–9`) confirm that DPSL remains competitive across a broad range of values rather than being sensitive to a narrow optimum.
>
> This robustness extends beyond MoEs. In our new unsupervised clustering experiments (our response to reviewer's next point - `Section 2.3` in the revised manuscript), we found that moderate priors around $\alpha = 1$ consistently work well, and even deliberately misspecified priors yield gains over baselines. This confirms that DPSL provides beneficial inductive bias without requiring fine-grained tuning.
>
> **Summary of Practical Guidelines (`Section 3.5`):**
>
> 1.  **Default:** Use symmetric $\alpha=1$ for all experts to achieve balanced specialization (our default across most models).
> 2.  **Small Models:** Optionally reduce to $\alpha=0.75$ for smaller models (<2B parameters) to encourage stronger specialization. Larger models are stable across $\alpha \in [0.75, 1.5]$.
> 3.  **Asymmetric Needs:** For specific preferences (e.g., modality imbalance), start from the base value and add $\approx 0.5$ to the preferred component (e.g., $\alpha = (1.5, 1, 1)$) to softly bias probability mass.
> 4.  **Weighting:** We find $\lambda = 0.01$ to be a robust default for the loss weight across all settings, with tuning effort comparable to standard MoE load-balancing coefficients.
>
>
> **W2)** We thank the reviewer for this suggestion. To demonstrate DPSL's generality beyond MoE routing, we applied it to **unsupervised deep clustering**, a canonical task involving categorical output distributions.
>
> We integrated DPSL into established clustering baselines, SwAV (Caron et al., 2020) and SeCu (Yan et al., 2023), using synthetic datasets with imbalanced cluster sizes (ratios 4:1:1, 5:2:1, and 5:3:1). While standard methods often favor balanced partitions, real-world clusters are frequently asymmetric. We used DPSL to impose *moderate asymmetric Dirichlet priors* on the cluster assignments.
>
> **Results:** As detailed in the new `Section 2.3`, DPSL yielded substantial accuracy gains across all regimes for both SwAV ($4.6–14.9$\%) and SeCu ($5.9–12.6$\%), as shown below (mean $\pm$ standard error across three random seeds):
>
> | Setting | SwAV | SwAV+DPSL | SeCu | SeCu+DPSL |
> | :--- | :--- | :--- | :--- | :--- |
> | Non-overlapping | 84.4 ± 0.8\% | **99.4 ± 0.1\%** (+14.9\%) | 88.4 ± 11.4\% | **94.3 ± 5.5\%** (+5.9\%) |
> | Overlapping | 86.7 ± 1.1\% | **94.1 ± 0.2\%** (+7.4\%) | 70.7 ± 3.8\% | **83.3 ± 1.7\%** (+12.6\%) |
> | Elongated | 87.7 ± 0.7\% | **92.3 ± 0.5\%** (+4.6\%) | 76.8 ± 7.1\% | **87.7 ± 3.9\%** (+10.8\%) |
>
> **Robustness to Prior Misspecification:** To test robustness, we also used a fixed, sub-optimal prior ($\alpha = (1.5, 1, 1)$) across all settings. As shown in `Appendix B` `Table 4`, DPSL still consistently improved performance over the baselines, confirming it serves as a beneficial structural prior even when the target is only approximately known.
>
> Full experimental details (`section 2.3` and `Appendix B`), visualizations of the qualitative impact on cluster shapes (`Figure 5`), and the rationale for prior selection are provided in the revised manuscript. These results validate DPSL as a general-purpose regularizer for shaping categorical distributions.
>
> **Q1 \& Q2)** We thank the reviewer for this suggestion and spotting our oversight. We have unified the terminology to use "asymmetric priors" consistently and have corrected the labeling of the second-best values in Table 2.
>
> Please let us know if these clarifications and the additional results in the revised paper adequately address your concerns. We are happy to provide any further clarification.

---

### Official Review · Reviewer_2RyL · 2025-11-03

**Soundness:** 3
**Presentation:** 3
**Contribution:** 3
**Rating:** 4
**Confidence:** 4

**Summary:**

This paper addresses the challenge of poor expert specialization with upcycled MoE models, where pre-trained dense models are converted to (sparse) MoEs. The authors propose the use of Dirichlet priors in order to regularize router probability distributions, matching expert assignments to target Dirichlet priors. The method shows good results in providing fine-grained control over expert balance/specialization, with experiments on popular vision language models that demonstrate improvements over baseline approaches.

**Strengths:**

- the challenge adressed is well defined, and explained intuitively through visualizations

- the use of Dirichlet priors to shape routing distributions is well motivated

- the framework can be more general than the presented application of MoE routing (e.g. in cases where modules output categorical probability distributions). Symmetric and asymmetric priors are both integrated in the framework

- evaluation spans multiple backbones and MoE configurations, indicating broad applicability

- ablation studies support the strategies employed, and provide insights into the method's behavior

**Weaknesses:**

- While the application to MoE routing is novel, the methodological innovation is incremental. The paper could benefit from discussing fundamental differences from related work.

- while improvements are consistent, absolute improvements are often small (e.g., even less than 1% in some cases), which makes the improvement questionable at times given the additional computational overhead

- although the framework can be more generally applicable, this is not evidenced in the paper

- The paper could benefit from more in depth analysis; e.g. specialization evidence (Sec 3.6), it is not clearly demonstrated that experts actually learn different functions, and there is no analysis of what each expert specializes in.

-In Table 2, modality-specific experiments are shown that demonstrate task-specific specialization that may sometimes underperform symmetric priors which could be seen as contradicting the specialization narrative.

**Questions:**

please see above. Additionally,
- more evidence for expert specialization - are experts really learning different functions? (E.g. via activations)
- how does this approach compare to simpler approaches, e.g. entropy regularization
- When should asymmetric priors be used? Given task-specific priors underperform e.g. in Table 2
- How stable is the method with respect to parametrization? can some settings lead to collapse?
- Scaling/computational overhead when increasing number of experts?
- Make sure to describe well how Avg score in Table 1 is computed

---

> ### Author Response · Authors · 2025-11-22
> **Response to the Reviewer 2RyL (1/3)**
>
> We thank the reviewer for the thoughtful feedback and for recognizing our Dirichlet framework as a well-motivated, generalizable approach to expert routing. We also appreciate the positive assessment of our comprehensive evaluation and ablation studies. Below, we address the specific questions and concerns raised.
>
> **W1)** We respectfully disagree with the characterization of our methodological contribution as incremental. This view contrasts with the consensus of other reviewers who emphasized the novelty and significant control offered by our framework. For instance, **Tjkt** described DPSL as *"a clean approach that provides fine-grained control over router behavior,"* noting that it *"moves beyond simple heuristics like load balancing and allows practitioners to instill complex and desirable statistical properties into the routing mechanism."* Similarly, **6fqQ** highlighted that *"the method allows flexible control over the trade-off between expert balance and specialization by aligning router outputs with a tunable Dirichlet prior."* Fundamentally, DPSL is a general, theoretically grounded framework for shaping categorical output distributions via Dirichlet priors and is not tied to MoE routing in particular.
>
> In the revised manuscript, we clarify this generality and add an additional experiment in `Section 2.3`, where DPSL is applied in an **unsupervised clustering** setup, yielding substantial improvements in clustering quality. This further demonstrates that the core idea is methodologically novel beyond the specific upcycled-MoE application.
>
> Regarding the comparison to "simpler" approaches such as entropy regularization, first, entropy regularization acts *per token* and pushes each individual routing distribution toward high entropy, thereby discouraging confident assignments and typically shrinking all router outputs toward the center of the simplex. In contrast, DPSL matches the *batch-level* empirical distribution of scores to a target Dirichlet prior over experts, which allows the router to retain highly confident assignments at the token level while only *constraining the aggregate statistics over many tokens*. Empirically, we show in `Figure 3` and `Appendix D.5` that conventional regularizers (including load-balancing and z-loss) produce router probabilities tightly concentrated around the uniform-selection value, whereas DPSL yields broader, more diverse score distributions that better support specialization.
>
> Second, DPSL exposes explicit, interpretable control knobs via the Dirichlet concentration parameters. By varying the magnitude and asymmetry of the prior, practitioners can smoothly interpolate between uniform routing, confident but balanced specialization, and targeted specialization (e.g., modality-aware or task-aware), without changing the underlying architecture or adding hand-crafted heuristics. This level of control is not available with entropy penalties, load-balancing losses, or z-loss.
>
> We have expanded the related-work section to emphasize these conceptual differences and to clarify that existing MoE regularizers operate either on logits (z-loss) or on simple aggregate counts (load balancing), whereas DPSL directly regularizes the shape of the routing distribution through a principled probabilistic prior.
>
>
>
> **W2)** We thank the reviewer for raising this point. We believe the improvements are practically meaningful and the overhead is justified for several reasons.
>
> First, DPSL is the only method that yields the best average performance across *all* three backbones (Qwen2, Phi-3, Llama3.2) and both MoE configurations (2in4 and 8in16). In contrast, the "runner-up" varies significantly by setting: e.g., a method that is competitive on Phi-3 2in4 can lag behind DPSL by around 1–1.5 points on Qwen2 models. This suggests that DPSL is not tuned to a single favorable configuration but serves as a robust regularizer across architectures and granularities.
>
> Second, for vision-language models at this scale, consistent improvements of up to 1\% are substantial. Training pipelines for modern VLMs (like LLaVA) are already heavily optimized, making further gains challenging.
>
> Finally, regarding computational cost, we have significantly optimized our implementation on H100 GPUs, reducing the training overhead significantly (detailed breakdown provided in our response to `Question 5`). Furthermore, DPSL only adds this minor overhead during the warm-up phase and incurs *zero* additional cost at inference time. Given its robustness and the zero-cost deployment, we consider this a highly favorable trade-off for practitioners.

---

> > ### Author Response · Authors · 2025-11-22
> > **Response to the Reviewer 2RyL (2/3)**
> >
> > **W3)** We thank the reviewer for this suggestion. To demonstrate DPSL's generality beyond MoE routing, we applied it to *unsupervised deep clustering*, a canonical task involving categorical output distributions.
> >
> > We integrated DPSL into established clustering baselines, SwAV (Caron et al., 2020) and SeCu (Yan et al., 2023), using synthetic datasets with imbalanced cluster sizes (ratios 4:1:1, 5:2:1, and 5:3:1). While standard methods often favor balanced partitions, real-world clusters are frequently asymmetric. We used DPSL to impose *moderate asymmetric Dirichlet priors* on the cluster assignments.
> >
> > **Results:** As detailed in the new `Section 2.3`, DPSL yielded substantial accuracy gains across all regimes for both SwAV ($4.6–14.9$\%) and SeCu ($5.9–12.6$\%), as shown below (mean $\pm$ standard error across three random seeds):
> >
> > | Setting | SwAV | SwAV+DPSL | SeCu | SeCu+DPSL |
> > | :--- | :--- | :--- | :--- | :--- |
> > | Non-overlapping | 84.4 ± 0.8\% | **99.4 ± 0.1\%** (+14.9\%) | 88.4 ± 11.4\% | **94.3 ± 5.5\%** (+5.9\%) |
> > | Overlapping | 86.7 ± 1.1\% | **94.1 ± 0.2\%** (+7.4\%) | 70.7 ± 3.8\% | **83.3 ± 1.7\%** (+12.6\%) |
> > | Elongated | 87.7 ± 0.7\% | **92.3 ± 0.5\%** (+4.6\%) | 76.8 ± 7.1\% | **87.7 ± 3.9\%** (+10.8\%) |
> >
> > **Robustness to Prior Misspecification:** To test robustness, we also used a fixed, sub-optimal prior ($\alpha = (1.5, 1, 1)$) across all settings. As shown in `Appendix B` `Table 4`, DPSL still consistently improved performance over the baselines, confirming it serves as a beneficial structural prior even when the target is only approximately known.
> >
> > Full experimental details (`section 2.3` and `Appendix B`), visualizations of the qualitative impact on cluster shapes (`Figure 5`), and the rationale for prior selection are provided in the revised manuscript.
> >
> >
> > **W4)** We thank the reviewer for this suggestion. To address this, we now include an expert activation similarity analysis that directly addresses whether experts learn different functions (`Section 3.6` and `Appendix D.6`). Specifically, we compute pairwise cosine similarity between expert activation patterns and visualize the pairwise similarities across layers 3, 6, 9, and 12, on 50 randomly selected MMVet samples in `Figure 8`. The results show that DPSL maintains the lowest average activation similarity (0.39) compared to load-balancing loss (0.57) and router z-loss (0.59). The asymmetric similarity patterns for our method indicate that different expert pairs specialize in different ways, rather than learning uniform representations. This asymmetry is largely absent in baseline methods, which show more uniform similarity across all expert pairs.
> >
> >
> >
> > **W5)** We respectfully disagree that `Table 3` contradicts our narrative. DPSL's primary novelty is its ability to inject *any* desired prior into expert routing, providing a flexible tool to encode domain knowledge when beneficial. `Table 3` demonstrates this flexibility by showing DPSL can accommodate symmetric priors (general-purpose), modality-specific priors (vision vs. language), and task-specific priors, all within the same framework.
> >
> > Regarding the performance differences:
> >
> > *   **Modality-specific priors** are a natural fit for VLMs because vision and language tokens possess distinct, consistent structures. `Table 3` confirms this alignment leads to the best performance (36.18 avg).
> >
> > *   **Task-specific priors** require manually defining "tasks", which introduces potential sub-optimality. As noted in `Section 3.4`, the lower performance (35.35 avg) likely reflects the multifaceted nature of tasks and non-optimal task definitions rather than a fundamental limitation of the method. Asymmetric priors are most effective when the underlying domain structure is well-understood and correctly specified.
> >
> >
> > **Practical Deployment Scenarios:** Critically, this flexibility enables powerful edge deployment use cases where *system constraints*, not just pure accuracy, dictate the prior. For example, on a memory-constrained device deploying a 4in32 setting, where only 16 experts fit in fast DRAM:
> >
> > *   A 4-in-32 MoE can utilize a *task-specific prior* (e.g., for "coding" tasks) to bias routing toward a specific 16-expert subset.
> > *   This effectively induces 4-in-16 behavior for that high-priority task, allowing those 16 experts to be pinned in fast memory.
> > *   This can result in a significant boost in cache hit rates and throughput for critical tasks, trading a negligible amount of theoretical accuracy for substantial real-world latency gains.
> >
> > We are happy to clarify in the revision that specialized task priors are particularly valuable in scenarios governed by clear external structures or specific hardware constraints.

---

> > > ### Author Response · Authors · 2025-11-22
> > > **Response to the Reviewer 2RyL (3/3)**
> > >
> > > **Q1)** Addressed in our response to **W4**
> > >
> > > **Q2)** Addressed in our response to **W1**
> > >
> > > **Q3)** Addressed in our response to  **W5**
> > >
> > > **Q4)** We never observed expert collapse or diverging runs across all settings, backbones, and configurations. DPSL is highly stable and inherently yields balanced expert loads (as analyzed in `Appendix D.5` \& `Table 12`), eliminating the need for auxiliary load-balancing losses.
> > >
> > > Regarding parametrization sensitivity, `Tables 8` \& `9` (Appendix D.1) show DPSL is robust across concentration parameter $\alpha$ values ranging from 0.75 to 1.5. Similarly, the regularization weight $\lambda$ ablation (`Table 11`, `Appendix D.3`) demonstrates consistent performance across the range [0.001, 0.1]. We observed zero diverging runs across all backbones and hyperparameters.
> > >
> > > **Q5)** We appreciate the reviewer’s detailed inquiry into these costs. We have included a detailed analysis of the computational costs in `Appendix C.5` of the revised manuscript, which we summarize below. Please refer to `Table 6` (standard training) and `Table 7` (DeepSpeed Zero-3) in that section for the specific profiling data. Our deeper analysis reveals that the overhead is minor, constant with respect to model size, and primarily driven by temporary implementation constraints rather than mathematical complexity.
> > >
> > > **Optimized Implementation \& Profiling:** The primary cost in our initial code was the CPU-GPU transfer required to compute Beta CDFs via NumPy/SciPy. For this rebuttal, we optimized the implementation to minimize these transfers and redundant reshaping. As shown in `Table 6` and `7` in `Appendix C.5`, the computational cost of DPSL on a Qwen2.5-0.5B model is as follows:
> > > *   **Standard Training:** 7-12\% overhead depending on expert count.
> > > *   **DeepSpeed Zero-3:** Overhead drops to 1-5\%.
> > >
> > > For larger models (Qwen2.5-1.5B), the relative gap shrinks further to negligible levels, as the CDF computation cost is constant while the model's forward/backward pass cost grows.
> > >
> > > **Vanishing Cost with Native GPU Support:** Crucially, the current overhead is an artifact of using SciPy for CDFs. PyTorch is actively adding native support for special functions (e.g., `torch.special.betainc`), and other frameworks like JAX/TensorFlow already support this on GPU. Once fully native, the overhead will effectively vanish, as the operation is faster and avoids CPU-GPU transfer overhead.
> > >
> > > **End-to-End Impact:**
> > > DPSL is active only during the warm-up phase (Stage II), which accounts for ~40\% of our training pipeline. Using DeepSpeed, a 1-5\% overhead in this stage translates to a trivial **0.4-2\% increase in total wall-clock time**. There is *zero* cost at inference time.
> > >
> > > **Q6)** Thank you. We added to the manuscript:
> > > "We also report the average accuracy (unweighted mean across seven metrics) for MME-P and MME-C, the scores were normalized over the maximal possible scores (2000 and 800, respectively)."
> > >
> > > Please let us know if these clarifications and the additional results in the revised paper adequately address your concerns. We are happy to provide any further clarification.

---

### Author Response · Authors · 2025-11-22
**Global Response to All Reviewers**

We thank the reviewers for their valuable feedback and thorough evaluations, which allowed us to significantly strengthen the paper. We have uploaded an updated version of the manuscript. Below, we summarize the main points raised and outline the additions and revisions made to the paper.

### **Novelty, Motivation, and General Applicability**
We are encouraged that reviewers found our core challenge, poor expert specialization in upcycled models, to be **"well-defined and explained intuitively" (Reviewer 2RyL)** and our proposed solution, Dirichlet-Prior Shaping Loss (DPSL), to be **"well-motivated" (Reviewer 2RyL)** and **"aligned well"** with this motivation **(Reviewer ZHna)**. Reviewer **Tjkt** particularly highlighted DPSL as a **"clean approach"** that moves beyond simple heuristics like load balancing to offer **"fine-grained control over router behavior,"** allowing practitioners to instill **"complex and desirable statistical properties."**

While acknowledging these strengths, reviewers (**2RyL, 6fqQ, Tjkt**) requested further evidence of DPSL’s generality beyond MoE routing.
*   **Improvement:** To demonstrate that DPSL is a general-purpose tool for shaping categorical distributions, we added `Section 2.3` and `Appendix B` detailing its application to **unsupervised deep clustering**. By integrating DPSL into established standard baselines (SwAV and SeCu) on asymmetric cluster data, we achieved substantial accuracy gains (up to **14.9%**), proving its utility as a broad statistical framework beyond the MoE context.

### **Experimental Rigor and Expert Specialization**
Reviewers commended our **"comprehensive experimental evaluation"** spanning multiple backbones (Qwen2, Phi3, Llama3.2), MoE configurations, and benchmarks (**2RyL**, **Tjkt**). Reviewer **6fqQ** noted that **"experimental results demonstrate its strong performance,"** while Reviewer **2RyL** appreciated that our ablation studies **"provide insights into the method's behavior."** Reviewer **Tjkt** specifically praised our router analysis for offering **"valuable qualitative insight"** into why DPSL encourages more confident routing.

Addressing questions (**2RyL, Tjkt**) about whether experts truly learn distinct functions:
*   **Improvement:** We conducted a new analysis in `Section 3.6` and `Appendix D.6`, measuring pairwise cosine similarities between expert activations. As visualized in `Figure 8`, DPSL maintains significantly lower similarity (**0.39**) compared to load-balancing (**0.57**) and z-loss (**0.59**), confirming that our method successfully drives **diverse expert specialization**.

### **Computational Efficiency and Practical Guidelines**
Reviewers appreciated that **"the method allows flexible control" (Tjkt, 2RyL)** over the trade-off between balance and specialization. Addressing concerns (**2RyL, Tjkt, ZHna**) regarding computational overhead and hyperparameter sensitivity:
*   **Improvement (Training Efficiency):** We optimized our implementation to minimize CPU-GPU transfers. New profiling in `Appendix C.5` confirms the training overhead is now negligible (**1–5%** under DeepSpeed Zero-3 during *stage II*, translating to a mere **0.4–2%** wall-clock increase), with **zero cost at inference**, reinforcing DPSL as a practical, low-cost solution.
*   **Improvement (Guidelines):** To support the observation that DPSL is a powerful tool for control, we added `Section 3.5` ("Practical guidelines"). We clarify that the Dirichlet parameter $\alpha$ is a robust, interpretable prior rather than a sensitive hyperparameter, recommending $\alpha=1$ as a universal starting point that performs stably across architectures ($\alpha \in [0.75, 1.5]$). Crucially, we find that this guideline holds across domains: moderate priors ($\alpha \approx 1$) consistently yielded gains in our new unsupervised clustering experiments as well, confirming that these settings serve as a robust default for shaping categorical distributions generally.

### **Summary of Additions and Revisions**
We have incorporated the following changes into the revised manuscript:
*   **New Section 2.3 & Appendix B:** Unsupervised deep clustering experiments demonstrating general applicability.
*   **New Section 3.5:** Practical guidelines for $\alpha$ selection.
*   **New Section 3.6 & Appendix D.6:** Expert activation similarity analysis (`Figure 8`) proving distinct expert specialization.
*   **Appendix C.5:** Detailed profiling of computational overhead (`Tables 6 & 7`).
*   **Related Work:** Expanded to clarify how DPSL differs from existing methods (entropy regularization, load-balancing, z-loss).
*   **Clarifications:** Explicitly stated the conjugate prior relationship of the Dirichlet distribution (`Section 2.1`) and clarified the calculation of average scores in `Table 2`.
*   **Terminology:** Unified the use of "asymmetric priors" throughout the text.
*   **Corrections:** Corrected the labeling of second-best results in `Table 2`.

---

### Meta-Review · Area_Chair_khv1 · 2026-01-07

**Summary:**

In their paper, the authors introduce Dirichlet distributions as a principled prior to improve expert specialization in sparse upcycling settings, where all experts are identically initialized and conventional training strategies lead to indifferent routing behavior. The core contribution of the work is to leverage a Dirichlet prior over routing probabilities and to introduce a Dirichlet-Prior Shaping Loss (DPSL) that regularizes the empirical routing distribution to match the target prior distribution.

(1) *Sensitivity to and hyperparameters*: The major concern raised by multiple reviewers (including Reviewer 6fqQ, Tjkt and ZHna) is that the proposed method introduces additional hyperparameters whose selection is insufficiently justified. In particular, the choice of the Dirichlet concentration parameters and the regularization weight requires tuning but the paper provides only heuristic guidelines rather than principled selection criteria. As several reviewers point out, the method largely reduces to selecting an appropriate prior distribution and enforcing the router to match it, without providing a sufficiently strong theoretical justification for this choice, which makes the contribution appear incremental rather than fundamental. Moreover, the need to tune these parameters across different backbones, datasets, and tasks significantly limits the practicality of the approach. Given the high computational cost of training and upcycling large MoE models often requiring multiple days on H100-class hardware, as reported by the authors in their experiments, methods that require extensive hyperparameter exploration are difficult to deploy in realistic settings.

(2) *Lack of formalization of the balance–specialization trade-off*: A second fundamental weakness is that DPSL does not clearly formalize the trade-off between expert balance and expert specialization, despite this being central to the paper's motivation. While the authors qualitatively argue that symmetric Dirichlet priors encourage balanced expert usage and asymmetric priors promote specialization, this trade-off is not captured through an explicit objective, constraint, or metric. Balance and specialization are instead evaluated post hoc, making it unclear how practitioners should systematically choose priors that achieve both desirable load balance and meaningful specialization. Reviewers also note that task-specific asymmetric priors sometimes underperform symmetric priors, which raises questions about whether strong specialization is always a suitable inductive bias for multi-faceted tasks and highlights the absence of a principled framework to navigate this trade-off.

The authors have made an effort to improve the presentation and address reviewer concerns, including extending DPSL to unsupervised deep clustering to demonstrate broader applicability. However, these additions do not fundamentally resolve the core issues of hyperparameter sensitivity and the lack of a formal balance-specialization framework. As a result, I am inclined toward rejection and encourage the authors to address these weaknesses more fundamentally, potentially through adaptive or data-driven prior selection and a clearer formalization of the balance–specialization trade-off in future submissions.

**Reviewer Concerns:**

Please refer to the summary.

**Reviewer Scores:**

Please refer to the summary.

---

### Decision · Program_Chairs · 2026-01-26

Reject